# Occupational and geographical differentials in financial protection against healthcare out-of-pocket payments in Nepal: Evidence for universal health coverage

**Vishnu Prasad Sapkota**[1]*, **Umesh Prasad Bhusal**[2], **Govinda Prasad Adhikari**[3]

**1** Department of Economics, Nepal Commerce Campus, Tribhuvan University, Kathmandu, Nepal, **2** Public Health and Social Protection Professional, Kathmandu, Nepal, **3** Department of Population Studies, Padma Kanya Multiple Campus, Tribhuvan University, Kathmandu, Nepal

* vishnu.sapkota@ncc.tu.edu.np

## Abstract

### Background

The low-and middle-income countries, including Nepal, aim to address the financial hardship against healthcare out-of-pocket (OOP) payments through various health financing reforms, for example, risk-pooling arrangements that cover different occupations. World Health Organization (WHO) has recommended member states to establish pooling arrangements so that the financial risks owing to health uncertainty can be spread across population. This study aims to analyse the situation of financial protection across occupations and geography using nationally representative annual household survey (AHS) in Nepal.

### Methods

We measured catastrophic health expenditure (CHE) due to OOP using two popular approaches—budget share and capacity-to-pay, and impoverishment impact at absolute and relative poverty lines. This study is the first of its kind from south-east Asia to analyse disaggregated estimates of financial protection across occupations and geography. The inequality in financial risk protection was measured using concentration index. Data were extracted from AHS 2014–15 –a cross-sectional survey that used standard consumption measurement tool (COICOP) and International Standard Classification of Occupations (ISCO).

### Results

We found a CHE of 10.7% at 10% threshold and 5.2% at 40% threshold among households belonging to agricultural workers. The corresponding figures were 10% and 4.8% among 'plant operators and craft workers'. Impoverishment impact was also higher among these households at all poverty lines. In addition, CHE was higher among unemployed households. A negative concentration index was observed for CHE and impoverishment

**Data availability statement:** All data used in this study are publicly available and can be obtained upon request from the National Statistics office of the Government of Nepal (https://microdata.nsonepal.gov.np/index.php/home).

**Funding:** Enter: The author(s) received no specific funding for this work.

**Competing interests:** The authors have declared that no competing interests exist.

impact among agricultural workers and 'plant operators and craft workers'. In rural areas, we found a CHE of 11.5% at 10% threshold and a high impoverishment impact. Across provinces, CHE was 12% in Madhesh and 14.3% in Lumbini at 10% threshold, and impoverishment impact was 1.9% in Madhesh, Karnali and Sudurpachim at US $1.90 a day poverty line.

## Conclusion

Households belonging to informal occupations were more prone to CHE and impoverishment impact due to healthcare OOP payments. Impoverishment impact was disproportionately higher among elementary occupations, agricultural workers, and 'plant operators and craft workers'. Similarly, the study found a wide urban/rural and provincial gap in financial protection. The results can be useful to policymakers engaged in designing health-financing reforms to make progress toward UHC.

## Introduction

Financial protection is an important dimension of Universal Health Coverage (UHC). It measures the extent to which the expenses while using healthcare by households comes at the cost of compromised living standards [1]. It is an important indicator of Sustainable Development Goals (SDG), and often used to gauge how well countries protect their citizens from financial hardship while using needed health services [2]. To make progress in this dimension, the World Health Organization (WHO) recommends suitable health financing instruments for the member countries [3]. Many low-and middle-income countries (LMIC) have been undertaking these reforms for UHC in the last few decades [1, 4]–for example, countries like Thailand, Ghana, Tanzania, and Indonesia have successfully structured the health financing arrangements for the occupation groups in the formal and informal sector of the economy [1, 4, 5]. Similarly, Turkey introduced a non-contributory health insurance scheme named Green Card in 1992 (further revised in 2005 to include outpatient services and pharmaceuticals) to provide financial protection to poor households that were previously not covered by formal health insurance scheme [6]. Health system and financing reforms under Health Transformation Program in Turkey are also successful in reducing inequity in health outcomes and making progress toward universal coverage [6, 7]. The pooling arrangements in these countries are broadly categorized as *contributory* schemes for the formal occupations and *tax-funded* arrangements for informal occupations and indigents [1, 2]. Therefore, the size and coverage of risk pooling schemes among occupation groups of the formal and informal sector of the economy determine country's progress in the financial protection and UHC a predominant way [1, 4]. Health financing reforms, in addition, creates incentives for availability and quality of health services through pre-agreed package and standards of healthcare between risk-pooling agency (purchaser) and healthcare provider [8].

Globally, informal sector occupies a major share of the population belonging to different occupations– 60% of the world's population have occupation in the informal sector. The corresponding figures for Asia and Africa are 68% and 86%, respectively [9]. Nepal Labour Force Survey (2017–18) reports that 62% people work in informal occupations, and nearly three-fourths of them are in the non-agricultural sectors [10]. Across geography, the distribution of workers in the informal sector is uneven [10]. International Standard Classification of Occupations (ISCO-08) classifies occupations that are spread across formal and informal sectors of the economy into a defined set of groups based on the nature of job, skill level, task, and duties undertaken in the occupations [11].

The ISCO classification, besides serving as a classification system for coherent labour sector management and reporting, also reveals immanent-health-risks, healthcare use, and financial hardship faced by workers. Across the occupation groups, the nature of tasks, duties, and working conditions in the workplace exposes them to the inherent hazards and generates incentives for a given pattern of health behaviours [12–16] that derives differential coverage of health services and social health protection schemes in LMICs [9, 17, 18]. Besides this, occupations directly affect the earning potential and financial status that is closely associated with the use of healthcare when needed [10, 19]. Similarly, the informal sector is not fully subjected to national labour legislation, income taxation, and social protection entitlements [20]. For example, the Government of Nepal (GoN) aims to cover the different groups of occupations through different social security arrangements. Employees Provident Fund covers civil servants and formal sector workers, and Social Security Fund covers private sector workers. Health Insurance Board focuses on the informal sector with the federal government subsidising the premium for (ultra) poor households [5]. However, these schemes together cover only less than 20% of the total population of Nepal [21–23], leaving a great majority of the population outside the risk pools. These factors together bring a pattern of financial hardship among the occupations not covered by the existing risk-pooling arrangements [24–26], and more seriously to those in the lower-income spectrum and occupations engaged in informal sector. These arrangements impose explicit and implicit non-cost rationing [5, 27, 28], and consequently increase the use of private healthcare which is likely to aggravate financial protection.

There are two commonly used approaches to measure the extent of financial hardship and financial protection in healthcare: (i) catastrophic health expenditure (CHE) and (ii) impoverishment due to healthcare out-of-pocket (OOP) payments [2, 29]. CHE is an SDG-3.8.2 indicator and is used to monitor financial hardship due to healthcare OOP payments. According to a recent estimate by Wang and colleagues, 10.7% (at 10% threshold) and 2.4% (at 25% threshold) of the population from Nepal experienced CHE [30]. Similarly, healthcare OOP pushes 1.67% and 3.68% Nepalese population below the poverty line of US $1.90 and US $3.20 per capita per day (pcpd), respectively. A study by Pyakurel et al. has reported a higher level of healthcare OOP, and a significant financial hardship among industrial workers seeking healthcare in Eastern Nepal [24]. While these figures are relevant at the national level, we found very few studies with disaggregated analyses of financial protection across occupations and geography in the national and international scenario.

In this context, our study aims to examine the financial protection statistics, both levels and distribution, across occupation and geography, using the nationally representative Annual Household Survey (AHS). The results of this study can be useful in several ways. First, the GoN through social protection schemes targets the population in different occupations. Therefore, this study will provide baseline, disaggregated estimates of financial protection to monitor the performance of existing risk-pooling arrangements. Second, the financial protection of occupations engaged in informal sectors, that occupy the greatest share (62%) [10], is particularly unknown. The coverage of these groups in existing social health protection schemes is also poor [21]. Having evidence on financial protection can be useful to policy makers to devise strategies, garner budgetary requirements and monitor effectiveness of the policies over time. Third, rural-urban and provincial disaggregation is important in terms of the country's federal structure since provincial and local planners and policymakers have significant budgetary discretion in devising strategies to improve financial protection. Fourth, occupational classification indicators move together with the size and composition of the economy, which is an important precondition for progress towards UHC [4, 31]. Fifth, the mutually exclusive occupation groups cover workers falling on a range in a socioeconomic spectrum; therefore, it is relevant to check, within each occupation group, whether the

financial protection statistics are concentrated more among the rich or poor. This piece of evidence can be useful in devising strategies to bridge the rich-poor gap in financial protection within each occupation group, an important aim of UHC.

## Materials and method

The study aims to analyse the financial protection status across occupations and geography using Annual Household Survey (AHS) 2014–15 –the most recent nationally representative cross-sectional survey that covers information on both the consumption expenditure and labour market. This study is the first of its kind from south-east Asia to analyse disaggregated estimates of financial protection across occupations and geography using standard methods. Next, we describe measurement of financial protection, the variables, and method of analysis.

### Financial protection

Financial protection against healthcare OOP is measured globally using two indicators: (i) CHE associated with healthcare OOP payments that reduce people's ability to allocate resources for other essential items and (ii) impoverishing health expenditure that push (or further push) people below the poverty lines [29]. These indicators together capture the impact of privately financed healthcare on household welfare. That is why these indicators provide the economic value of reducing uncertainty related to large healthcare OOP payments [32, 33]. The measurement of these indicators requires: (i) definition of household resources available-to-pay for healthcare (a measure of consumption), (ii) definition and measurement of healthcare OOP payments, and (iii) definition of CHE thresholds and poverty lines [34].

**Definition of household resources available-to-pay.** In literature, there are at least four alternative approaches to define household resources available-to-pay for healthcare:

i. *budget share approach*–also known as SDGs 3.8.2 approach—defines healthcare OOP expenses as CHE when they exceed a given percentage (10% or 25%) of household total consumption [2];

ii. *capacity-to-pay approach (CTP)*–also known as WHO-CTP approach—involves deduction of food expenditures from the *total consumption* explained in the (i) definition and uses 40% threshold to define CHE [35];

iii. *expanded-CTP approach* involves deduction of food, rent, and utility expenditures [36], or a multiplier of poverty lines [37] from the total consumption explained in the (i) definition.

The argument behind deducing a certain amount from the total consumption in (ii) and (iii) definition is to capture a better measure of a household's ability or capacity-to-pay healthcare OOP payments [38]. With the (i) definition, the CHE is usually less concentrated among "poor people" (or more concentrated among "rich people"). In order to address this bias at lower tail of income distribution, the CTP approach is used that records a higher incidence of CHE spending among the poor than the budget share approach. A comparative analysis of these two approaches provides a balanced understanding of CHE [39]. We used the first two approaches to measure financial protection indicators so that the results will be comparable to the available literature including the global reports published by WHO and World Bank [2]. The AHS used the Classification of Individual Consumption by Purpose (COICOP) to classify household consumption expenditures into 12 broad categories. The categories cover food (COICOP-1 and 11), and non-food consumption expenditures (COICOP 2–12) that includes tobacco and alcohol, clothing, housing, furnishing, transportation, communication, entertainment, health, education and other expenses. The details about the components of consumption are available in the AHS report [40].

**Measure of healthcare OOP.** The survey measured healthcare OOP using section six of the COICOP based consumption module. We disaggregated the healthcare OOP payment across four broad categories: (i) expenses on medicine, (ii) expenses on OPD/ambulatory care, (iii) expenses on hospital/nursing care, and (iv) expenses on medical equipment. These categories are consistent with the healthcare expenditure categories used by the recent global monitoring report on UHC and financial protection [2].

**CHE thresholds and poverty lines.** The thresholds provide the idea about household balance available to spend on other essential items after spending on healthcare [32, 39, 41]. The earlier studies have used different thresholds (commonly 10%, 25% and 40%), depending on the definition of household resources [2, 29, 42]. Consistent with the available literature, we presented the results of CHE at 10% and 25% thresholds with budget share approach, and at the 40% threshold using CTP approach. Similarly, we used two popular approaches, namely absolute international poverty lines of US $1.90 per capita per day and $3.20 per capita per day and the relative poverty line of 50% and 60% of median daily per capita consumption to measure impoverishment impact of healthcare OOP. These lines are widely used in the analysis of financial protection in literature [2, 29, 42]. We converted international poverty lines into national values using purchasing power parity (PPP) for September 2020.

Using total consumption ($c$), healthcare OOP ($oop$) and catastrophic threshold ($\tau$), CHE was estimated using Eq 1.

$$CHE = \frac{\Sigma_h hhsize_h \times w_h I\left(\dfrac{oop_h}{c_h} \geq \tau\right)}{\Sigma_h hhsize_h \times w_h} \tag{1}$$

where, $w$ is household weight and *hhsize* is the household size.

Similarly, impoverishment impact of healthcare OOP was estimated using Eq 2.

$$IMP = \frac{\Sigma_h hhsize_h \times w_h \left\{I\left((c_h - oop_h) \leq PL\right) - I\left(c_h \leq PL\right)\right\}}{\Sigma_h hhsize_h \times w_h} \tag{2}$$

We disaggregated the financial protection indicators across the household occupations, occupation sector, geography, and socioeconomic variables.

## Independent variables

We used variables related to occupations, sector of occupation, geography, and socioeconomic status as independent variables.

**Occupational classification.** We used 10 groups of occupations defined by Nepal Standard Classification of Occupation (NSCO) [40, 43]. The information on occupations was available for each member of the household. We further defined occupation of household into the six occupational categories based on the following sequential steps (Table 1).

**Occupation sector.** We classified households broadly into one of the following four sectors if at least one member had occupation in one of the categories: (i) salaried in non-agriculture, (ii) salaried in agriculture, (iii) self-employed in non-agriculture, and (iv) self-employed in agriculture.

In case where members from the same household were employed in a different sector, we considered household's major source of income as the household sector of occupation.

**Geographical variables.** We included province and area of residence (urban/rural) to explain the geographical variation in financial protection. Since this survey was conducted

**Table 1. Reclassification of NSCO in six categories.**

| Step | Household occupation | NSCO groups for each member |
|------|---------------------|------------------------------|
| 1 | Managers and professionals | Armed force (NSCO: 0), manager (NSCO: 1), professional (NSCO: 2), associate professional (NSCO: 3) |
| 2 | Clerical and support workers | Clerical support (NSCO: 4), and service and sales support (NSCO: 5) |
| 3 | Agriculture | Agriculture and forestry (NSCO: 6) |
| 4 | Plant operators and craft workers | Craft and related trade (NSCO: 7), and plant and machine operations (NSCO: 8). |
| 5 | Elementary occupations | Other elementary occupations (NSCO: 9). |
| 6 | Unemployed | If household does not report the occupation in any of the NSCO categories |

Note: If household members have occupation in more than one of the six categories, we classified the households based on the above-explained order. Abbreviation: NSCO = Nepal Standard Classification of Occupation

before the federalization of Nepal in 2017, we used district and municipality codes available in the dataset to prepare the province variable that corresponded to seven provinces of Nepal's federal structure.

**Socioeconomic variables.** Nepal Labor Force Survey (NLFS)-2019 reports that household level distribution of occupations is closely associated with important socioeconomic variables such as gender of household head, household head's education and the overall economic status of the household [10]. Therefore, we included sex, education of household head, and consumption quintile as socioeconomic variables to further explain the observed heterogeneity in financial protection statistics across occupations and geography.

## Measure of inequality

We calculated concentration index in its relative form for financial protection statistics across socioeconomic, geography, and occupational characteristics using the approach suggested by Kakwani and Wagstaff [44] using *decomp* package in R [45].

## Data source

We used the data from AHS 2014–15 conducted by Central Bureau of Statistics (CBS), Nepal. The purpose of the survey was to provide updated estimations on socioeconomic indicators, consumption, and labour force. This survey used a sampling frame of the National Population Census 2011, and covered 288 Primary Sampling Unit (PSU) (143 in urban and 145 in rural) and 4320 households (2145 from urban PSU and 2175 from rural PSU). The questionnaire contained household and individual level information. We used information from Sections 1, 3, 4, and 5 of the questionnaires in [40] that covered socio-demographic variables, the food and non-food consumption and healthcare OOP payments, and information on economic activities and occupations, respectively.

## Ethical approval and consent to participate

We based this study on a publicly available dataset, and the survey questionnaire is also available in the reports [40]. The permission to access and use this dataset was obtained from Central Bureau of Statistics (CBS), Government of Nepal. The protocol for the survey was approved by the Central Bureau of Statistics (CBS) based on the Statistical Act (1958) in September 2018. This Act enables CBS to implement surveys as per the Government of Nepal's ethics protocol without involving an institutional review board (IRB) approval. So, all

procedures were performed in accordance with relevant guidelines. During the data collection, verbal consent was obtained from each respondent after a thorough introduction of the survey. All respondents were briefed about the voluntary nature of participation. Participants were assured anonymity and confidentiality of shared information.

## Results

Table 2 shows the background characteristics of the households based on the occupational classification. We found 10.3% and 47.4% of households in formal occupation ('managers and professionals' and 'clerical and support workers') and agriculture workers, respectively. Regarding household head's characteristics, nearly three-fourths of households in all occupational groups had male household heads, except for unemployed household (35.3%). Regarding household head's education, we found 48.2% of household heads belonging to 'managers and professionals' and 43.9% of household heads belonging to 'clerical and support worker'. Differences in occupation groups were also observed across geography, most of the households with formal occupations and unemployed were from urban areas ('managers and professionals': 78.1%; and 'clerical and support workers': 71.7%; unemployed: 70.1%). On the contrary, the majority

**Table 2.** Household's occupation across background characteristics (N = 4320).

| Covariates | Managers and Professionals | Clerical and Support workers | Agricultural Workers | Plant Operators and Craft Workers | Elementary Occupations | Unemployed |
|---|---|---|---|---|---|---|
| **Sex of household head** | | | | | | |
| Male | 143(73) | 195(77.5) | 1447(70.6) | 562(84.4) | 670(76.1) | 98(35.3) |
| Female | 53(27) | 57(22.5) | 603(29.4) | 104(15.6) | 210(23.9) | 179(64.7) |
| **Education of household head** | | | | | | |
| Less than primary | 7(3.6) | 33(13.1) | 898(43.8) | 237(35.6) | 467(53) | 93(33.5) |
| Primary education | 19(9.7) | 69(27.5) | 650(31.7) | 273(41) | 290(33) | 62(22.3) |
| Secondary education | 94(48.2) | 110(43.9) | 387(18.9) | 129(19.4) | 93(10.6) | 103(37.2) |
| Higher education | 75(38.5) | 39(15.5) | 114(5.6) | 26(4) | 30(3.4) | 20(7.1) |
| **Area of residence** | | | | | | |
| Urban | 153(78.1) | 180(71.7) | 401(19.6) | 257(38.6) | 252(28.7) | 195(70.1) |
| Rural | 43(21.9) | 71(28.3) | 1648(80.4) | 409(61.4) | 628(71.3) | 83(29.9) |
| **Province** | | | | | | |
| Province 1 | 27(13.9) | 49(19.7) | 363(17.7) | 102(15.3) | 107(12.1) | 44(15.9) |
| Madhesh | 12(6.3) | 12(4.7) | 332(16.2) | 93(13.9) | 178(20.2) | 34(12.4) |
| Bagmati | 70(35.6) | 95(37.9) | 292(14.3) | 117(17.5) | 153(17.3) | 72(26.1) |
| Gandaki | 32(16.4) | 52(20.6) | 318(15.5) | 112(16.8) | 141(16.1) | 63(22.7) |
| Lumbini | 12(5.9) | 10(3.8) | 81(3.9) | 17(2.5) | 63(7.2) | 13(4.6) |
| Karnali | 37(18.8) | 28(11.2) | 460(22.4) | 177(26.6) | 148(16.8) | 44(15.8) |
| Sudurpaschim | 6(3) | 5(2) | 203(9.9) | 49(7.4) | 91(10.3) | 7(2.4) |
| **Consumption quintile** | | | | | | |
| Lowest (first) | 2(1) | 3(1.1) | 433(21.1) | 124(18.6) | 275(31.2) | 29(10.4) |
| Second | 10(5) | 12(4.8) | 485(23.7) | 117(17.5) | 211(24) | 30(10.6) |
| Middle | 13(6.7) | 26(10.4) | 460(22.4) | 151(22.7) | 171(19.4) | 45(16.2) |
| Fourth | 34(17.5) | 71(28.4) | 402(19.6) | 156(23.4) | 135(15.3) | 64(23) |
| Highest (fifth) | 137(69.9) | 139(55.4) | 270(13.2) | 118(17.7) | 89(10.1) | 110(39.7) |
| **Total** | **195(4.5)** | **251(5.8)** | **2050(47.4)** | **666(15.4)** | **880(20.4)** | **277(6.4)** |

Note: Numbers indicate frequency (percentage).

**Table 3.  Summary of consumption expenditure and healthcare OOP payments per capita (N = 4320).**

| Covariates | Consumption per capita per annum (NPR) | | | | Share of total consumption (%) | | |
|---|---|---|---|---|---|---|---|
| | Total | Food | Non-food | Healthcare OOP | Food | Non-food | Healthcare OOP |
| National | 64534 | 32231 | 24055 | 3134 | 0.594 | 0.406 | 0.043 |
| **Household occupations** | | | | | | | |
| Managers and professionals | 150390 | 52229 | 67590 | 4571 | 0.403 | 0.597 | 0.030 |
| Clerical and support workers | 118846 | 45687 | 48628 | 5029 | 0.453 | 0.547 | 0.042 |
| Agriculture | 53818 | 30240 | 19133 | 2810 | 0.625 | 0.375 | 0.045 |
| Plant operators and craft workers | 60144 | 31865 | 20791 | 2795 | 0.596 | 0.404 | 0.044 |
| Elementary Occupations | 47556 | 27722 | 15088 | 2010 | 0.645 | 0.355 | 0.039 |
| Unemployed | 98256 | 35801 | 43699 | 7181 | 0.460 | 0.540 | 0.049 |
| **Area of residence** | | | | | | | |
| Urban | 92565 | 38257 | 37521 | 4328 | 0.507 | 0.493 | 0.041 |
| Rural | 50536 | 29222 | 17330 | 2538 | 0.638 | 0.362 | 0.044 |
| **Consumption quintile** | | | | | | | |
| Lowest (First) | 21630 | 14959 | 5243 | 804 | 0.693 | 0.307 | 0.037 |
| Second | 34665 | 23182 | 9093 | 1324 | 0.670 | 0.330 | 0.038 |
| Middle | 46928 | 29808 | 13249 | 1920 | 0.636 | 0.364 | 0.041 |
| Fourth | 67122 | 38036 | 22263 | 3037 | 0.569 | 0.431 | 0.045 |
| Highest (Fifth) | 152447 | 55206 | 70489 | 8593 | 0.402 | 0.598 | 0.054 |

of households belonging to informal occupations were from rural areas. Regarding provinces, households belonging to informal occupations were distributed uniformly across provinces except for Lumbini and Sudurpaschim. Regarding consumption quintile, most households with formal occupation were at the upper tail of the consumption quintile (fourth and fifth quintile), and elementary occupations were at the lower tail of the consumption quintile (first, second and middle).

Table 3 provides a summary of consumption expenditure components across occupations, geography, and consumption quintiles. Total consumption expenditure was NPR 64534 per capita per annum, and food and non-food shares were 60% and 40% of the total consumption, respectively. The highest consumption expenditure was observed among the households with 'managers and professionals' (NPR 150390), and lowest for households with elementary occupation (NPR 47556). The food share was the lowest in the fifth consumption quintile (40.2%) and the highest in the first consumption quintile (69.3%), and this observation corresponds to Engel's law. Healthcare expenditure was NPR 3134 per capita per annum, and 4.3% of the total consumption. Healthcare expenditure share was the highest for households with the agricultural workers (4.5%) and the lowest for households with 'managers and professionals' (3.0%).

Fig 1 illustrates the components of healthcare OOP payments across geography, household occupations, and consumption quintiles. OOP payment on medicines occupied 75% share of the total healthcare OOP payments, followed by hospitalization (about 15%). Ambulatory services and health products together covered the rest of the share (<10%). Among the occupational groups, the share of hospitalization expenditure was about 20% among households with 'managers and professionals', and about 12% among unemployed households. OOP payment share on medicine was the highest among households in unemployed occupation and the lowest for households with 'managers and professionals'. Across the provinces, OOP payment for hospitalization was the highest in Bagmati and Gandaki, and the lowest in Madhesh. OOP payment for medicine was the highest in Sudurpaschim and Madhesh.

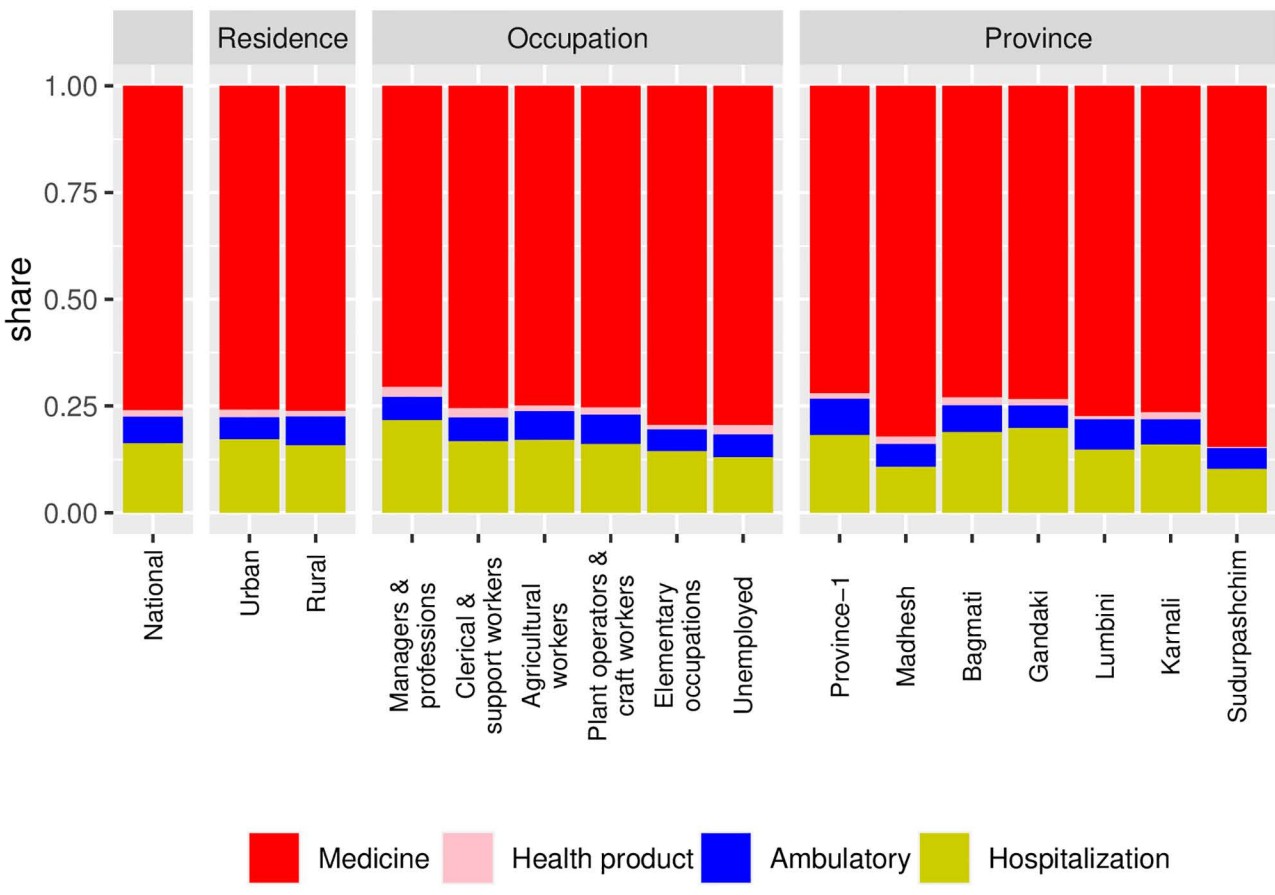

**Fig 1. Distribution of components of healthcare OOP share.**

Table 4 presents the incidence of CHE based on two different approaches. Using budget share approach, CHE was 10.1% and 2.3% at the threshold of 10% and 25%, respectively. Similarly, using the CTP approach, the CHE was 4.3% at 40% threshold.

We observed systematic differences in CHE across the occupations groups. Among the households belonging to 'managers and professionals', we observed 6% and 0.8% CHE at the thresholds of 10% and 25%, respectively using budget share approach, and 0.6% CHE using the CTP approach. Similar pattern was observed among salaried occupations in non-agricultural sector (Table 4). Among the households with 'clerical and support workers' and agricultural workers, we found differences in CHE only using the CTP approach (2.1% and 5.2%, respectively). Similar findings were observed among the households with salaried self-employed occupation in agriculture sector (Table 4). Among the households with elementary occupations, CHE was consistently less than the national average across all the definitions. Among the unemployed households, CHE was 12% and 3% at the thresholds of 10% and 25%, respectively using budget share approach, and 3.4% using the CTP approach.

Across the area of residence (rural/urban), we found a systematic difference in CHE using the CTP approach– 3% and 5% households were catastrophic in the urban and rural areas, respectively. Similarly, we found systematic differences in CHE across the provinces. In Madhesh and Lumbini provinces, CHE was, respectively, 12% and 14.3% at 10% threshold using budget share approach and 6% and 5% using CTP approach. In Gandaki and Karnali Provinces, similar pattern of CHE was observed using CTP approach. On the contrary, in Province

**Table 4. Catastrophic health expenditure across background characteristics (N = 4320).**

| Covariates | n | Catastrophic Health Expenditure (budget share approach) | | | | | Catastrophic Health Expenditure (CTP approach) | | |
|---|---|---|---|---|---|---|---|---|---|
| | | OOP share | Threshold (10%) | | Threshold (25%) | | OOP share | Threshold (40%) | |
| | | | value | 95% CI | value | 95% CI | | value | 95% CI |
| National | 4320 | 0.043 | 0.101 | 0.091–0.11 | 0.023 | 0.019–0.028 | 0.105 | 0.043 | 0.037–0.05 |
| **Household occupation** | | | | | | | | | |
| Managers and professionals | 195 | 0.030 | 0.063 | 0.032–0.094 | 0.008 | -0.001–0.018 | 0.050 | 0.006 | -0.002–0.015 |
| Clerical and support workers | 251 | 0.042 | 0.107 | 0.07–0.144 | 0.024 | 0.005–0.042 | 0.076 | 0.021 | 0.003–0.038 |
| Agriculture | 2050 | 0.045 | 0.107 | 0.092–0.121 | 0.025 | 0.018–0.033 | 0.114 | 0.052 | 0.041–0.063 |
| Plant operators and craft workers | 666 | 0.044 | 0.097 | 0.073–0.12 | 0.025 | 0.012–0.037 | 0.108 | 0.048 | 0.031–0.066 |
| Elementary occupations | 881 | 0.039 | 0.085 | 0.065–0.105 | 0.019 | 0.009–0.028 | 0.107 | 0.038 | 0.024–0.052 |
| Unemployed | 277 | 0.049 | 0.128 | 0.091–0.166 | 0.030 | 0.012–0.047 | 0.091 | 0.034 | 0.014–0.054 |
| **Employment status** | | | | | | | | | |
| Employed | 4043 | 0.043 | 0.099 | 0.089–0.109 | 0.023 | 0.018–0.028 | 0.106 | 0.044 | 0.037–0.051 |
| Unemployed | 277 | 0.049 | 0.128 | 0.091–0.166 | 0.030 | 0.012–0.047 | 0.091 | 0.034 | 0.014–0.054 |
| **Major sector of occupation** | | | | | | | | | |
| Salaried in non-agriculture | 449 | 0.035 | 0.087 | 0.061–0.113 | 0.009 | 0–0.017 | 0.080 | 0.019 | 0.005–0.032 |
| Salaried in agriculture | 92 | 0.051 | 0.129 | 0.055–0.204 | 0.042 | -0.005–0.09 | 0.151 | 0.072 | 0.011–0.132 |
| Self-employed in non-agriculture | 432 | 0.041 | 0.102 | 0.074–0.129 | 0.027 | 0.012–0.042 | 0.081 | 0.032 | 0.016–0.048 |
| Self-employed in agriculture | 3000 | 0.044 | 0.099 | 0.088–0.111 | 0.024 | 0.018–0.03 | 0.112 | 0.049 | 0.04–0.057 |
| **Area of residence** | | | | | | | | | |
| Urban | 1439 | 0.041 | 0.097 | 0.084–0.109 | 0.022 | 0.016–0.028 | 0.084 | 0.030 | 0.023–0.038 |
| Rural | 2881 | 0.044 | 0.103 | 0.09–0.116 | 0.024 | 0.017–0.03 | 0.115 | 0.050 | 0.041–0.059 |
| **Province** | | | | | | | | | |
| Province 1 | 693 | 0.038 | 0.091 | 0.068–0.113 | 0.016 | 0.006–0.025 | 0.094 | 0.028 | 0.015–0.041 |
| Madhesh | 662 | 0.050 | 0.120 | 0.094–0.147 | 0.026 | 0.013–0.039 | 0.138 | 0.060 | 0.04–0.079 |
| Bagmati | 801 | 0.042 | 0.100 | 0.079–0.122 | 0.030 | 0.017–0.042 | 0.093 | 0.041 | 0.026–0.056 |
| Gandaki | 717 | 0.047 | 0.106 | 0.082–0.129 | 0.028 | 0.015–0.041 | 0.106 | 0.050 | 0.033–0.067 |
| Lumbini | 194 | 0.049 | 0.143 | 0.094–0.192 | 0.037 | 0.01–0.063 | 0.119 | 0.050 | 0.019–0.081 |
| Karnali | 892 | 0.048 | 0.107 | 0.085–0.13 | 0.021 | 0.011–0.031 | 0.114 | 0.052 | 0.035–0.068 |
| Sudurpaschim | 360 | 0.022 | 0.035 | 0.016–0.054 | 0.008 | -0.001–0.016 | 0.061 | 0.011 | 0–0.022 |
| **Sex of household head** | | | | | | | | | |
| Male | 3113 | 0.044 | 0.103 | 0.092–0.115 | 0.024 | 0.018–0.03 | 0.109 | 0.048 | 0.04–0.057 |
| Female | 1207 | 0.041 | 0.094 | 0.076–0.111 | 0.021 | 0.013–0.03 | 0.094 | 0.031 | 0.02–0.041 |
| **Education of household head** | | | | | | | | | |
| Less than primary | 1735 | 0.043 | 0.097 | 0.082–0.112 | 0.020 | 0.013–0.027 | 0.118 | 0.045 | 0.034–0.055 |
| Primary Education | 1362 | 0.043 | 0.102 | 0.084–0.119 | 0.024 | 0.016–0.033 | 0.105 | 0.044 | 0.032–0.056 |
| Secondary Education | 920 | 0.044 | 0.107 | 0.086–0.127 | 0.030 | 0.018–0.042 | 0.090 | 0.047 | 0.032–0.062 |
| Higher Education | 304 | 0.039 | 0.100 | 0.068–0.132 | 0.017 | 0.004–0.03 | 0.079 | 0.022 | 0.006–0.038 |
| **Consumption quintile of household** | | | | | | | | | |
| Lowest (First) | 864 | 0.037 | 0.074 | 0.054–0.093 | 0.006 | 0–0.011 | 0.114 | 0.036 | 0.022–0.051 |
| Second | 864 | 0.038 | 0.080 | 0.061–0.099 | 0.014 | 0.005–0.022 | 0.109 | 0.040 | 0.026–0.054 |
| Middle | 864 | 0.041 | 0.100 | 0.078–0.122 | 0.022 | 0.011–0.033 | 0.107 | 0.037 | 0.023–0.051 |
| Fourth | 865 | 0.045 | 0.120 | 0.098–0.143 | 0.025 | 0.014–0.036 | 0.104 | 0.052 | 0.036–0.068 |
| Highest (Fifth) | 863 | 0.054 | 0.129 | 0.107–0.151 | 0.050 | 0.036–0.065 | 0.090 | 0.052 | 0.037–0.067 |

1 and Sudurpachhim province, CHE was 9.1% and 3.5%, respectively at 10% threshold using budget share approach.

Across sex of household heads, the analysis found no systematic difference in CHE using the budget share approach. However, using the CTP approach, the figure was 4.8% among male-headed households compared to 3.1% among the female-headed households. CHE was only 2.2% when the household head had higher education. Likewise, CHE using budget share approach was higher at higher consumption quintile but the increase was less pronounced using CTP approach.

Table 5 shows the impoverishing effect of healthcare OOP expressed as the share of population pushed below the poverty line after taking into account healthcare OOP. At the national level, 1.3% of population was pushed below the extreme poverty line of purchasing-power-parity US $1.90 per capita per day, and 1.5% and 2.4% population incurred impoverishing healthcare OOP using relative poverty lines of 50% and 60% of median consumption, respectively.

Across the occupations, we found heterogeneity in the impoverishment impact of healthcare OOP. Among the households belonging to 'managers and professionals' and 'clerical and support worker', the percentage of the population pushed below the poverty line due to healthcare OOP was nominal at both absolute and relative poverty lines. However, among households with occupation in agriculture, a relatively higher percentage of the population was pushed below the poverty line due to healthcare OOP– 1.3% and 4.3% at the poverty lines of $1.90 and $3.20 a day, respectively. In the same sub-group, when using relative poverty lines, 1.6% and 2.6% of the population were pushed below the poverty lines of 50% and 60% of median consumption, respectively. Households with 'plant operators and craft workers', also exhibited a higher degree of CHE and impoverishment impact (Table 5). A nominal percentage of households fall below the poverty line among the households belonging to the unemployed population. Similarly, we also found a relatively high impoverishment impact of healthcare OOP payments across salaried and self-employed occupations in the agriculture sector.

In the rural area, 1.7% and 4.0% of the households were impoverished due to healthcare OOP at the absolute poverty lines of US $1.90 and $3.20 a day, respectively. Across provinces, impoverishment due to healthcare OOP was the highest in Madhesh– 1.9% and 4.9%, at the poverty lines of US $1.90 and $3.2 a day, respectively. The Sudurpaschim and Karnali province also followed a similar pattern of impoverishment impact with all the definitions of poverty lines. However, the impoverishment impact was the lowest in Province 1 and Bagmati province, across all the definitions (Table 5).

There was a variation in impoverishing impact of healthcare OOP across socio-demographic characteristics. When the household head had only primary education, 2.0% and 3.7% of the households were pushed below the poverty lines of US $1.90 and $3.20 a day, respectively. On the contrary, when household heads had higher education, the impoverishment impact was nominal. Impoverishing effect of healthcare OOP was the highest in the first consumption quintile, nearly zero in the fourth and the fifth consumption quintile (Table 5).

Table 6 shows the share of population (those already below the poverty line) further pushed into poverty due to healthcare OOP expenditure. At the national level, an estimated 0.38% of the population was pushed further below the extreme poverty line of purchasing-power-parity US $1.90 per capita per day. At US $ 3.2 per capita per day, an estimated 1.25% of the population was pushed further down the poverty line due to healthcare OOP expenditure. Using relative poverty lines of 50% and 60% of median consumption estimated 0.39% and 0.68% of the population, respectively, was further pushed down the poverty due to healthcare OOP. Among household occupation groups, the percentage of the population pushed further down the poverty due to healthcare OOP was least for 'managers and professionals' and 'clerical and

**Table 5. Impoverishment impact of healthcare OOP (pushed below the poverty lines) across background characteristics (N = 4320).**

| Covariates | n | Absolute poverty lines (international poverty lines) | | | | Relative poverty lines (% consumption of median population) | | | |
| --- | --- | --- | --- | --- | --- | --- | --- | --- | --- |
| | | $1.90 per capita per day | | $3.20 per capita per day | | At 50% | | At 60% | |
| | | Value | 95% CI | Value | 95% CI | Value | 95% CI | Value | 95% CI |
| National | 4320 | 0.0129 | 0.009–0.017 | 0.0333 | 0.027–0.039 | 0.0153 | 0.011–0.02 | 0.0235 | 0.019–0.028 |
| **Household occupations** | | | | | | | | | |
| Managers and professionals | 195 | 0.0000 | – | 0.0073 | -0.007–0.022 | 0.0000 | – | 0.0000 | – |
| Clerical and support workers | 251 | 0.0054 | -0.005–0.016 | 0.0153 | -0.002–0.032 | 0.0112 | -0.004–0.027 | 0.0107 | -0.002–0.024 |
| Agriculture | 2050 | 0.0134 | 0.008–0.019 | 0.0430 | 0.033–0.053 | 0.0158 | 0.01–0.022 | 0.0256 | 0.018–0.033 |
| Plant operators and craft workers | 666 | 0.0168 | 0.005–0.028 | 0.0350 | 0.021–0.049 | 0.0166 | 0.005–0.028 | 0.0257 | 0.012–0.039 |
| Elementary occupations | 881 | 0.0174 | 0.007–0.027 | 0.0283 | 0.016–0.04 | 0.0220 | 0.011–0.033 | 0.0308 | 0.018–0.043 |
| Unemployed | 277 | 0.0025 | -0.002–0.007 | 0.0099 | -0.004–0.024 | 0.0025 | -0.002–0.007 | 0.0082 | -0.004–0.02 |
| **Employment status** | | | | | | | | | |
| Employed | 4043 | 0.0137 | 0.01–0.018 | 0.0349 | 0.029–0.041 | 0.0162 | 0.012–0.021 | 0.0245 | 0.019–0.03 |
| Unemployed | 277 | 0.0025 | -0.002–0.007 | 0.0099 | -0.004–0.024 | 0.0025 | -0.002–0.007 | 0.0082 | -0.004–0.02 |
| **Sector of occupation** | | | | | | | | | |
| Salaried in non-agriculture | 449 | 0.0098 | 0–0.02 | 0.0162 | 0.005–0.027 | 0.0080 | -0.001–0.017 | 0.0142 | 0.002–0.026 |
| Salaried in agriculture | 92 | 0.0000 | – | 0.0772 | 0.016–0.138 | 0.0142 | -0.013–0.042 | 0.0348 | -0.005–0.074 |
| Self-employed in non-agriculture | 432 | 0.0060 | -0.002–0.014 | 0.0138 | 0.002–0.025 | 0.0093 | -0.001–0.02 | 0.0109 | 0.001–0.02 |
| Self-employed in agriculture | 3000 | 0.0158 | 0.011–0.021 | 0.0395 | 0.032–0.047 | 0.0185 | 0.013–0.024 | 0.0277 | 0.021–0.034 |
| **Area of residence** | | | | | | | | | |
| Urban | 1439 | 0.0049 | 0.002–0.008 | 0.0221 | 0.016–0.028 | 0.0040 | 0.001–0.007 | 0.0150 | 0.01–0.02 |
| Rural | 2881 | 0.0170 | 0.011–0.022 | 0.0390 | 0.031–0.047 | 0.0210 | 0.015–0.027 | 0.0277 | 0.021–0.035 |
| **Province** | | | | | | | | | |
| Province 1 | 693 | 0.0140 | 0.004–0.024 | 0.0216 | 0.01–0.033 | 0.0195 | 0.008–0.031 | 0.0160 | 0.006–0.026 |
| Madhesh | 662 | 0.0186 | 0.007–0.03 | 0.0487 | 0.031–0.066 | 0.0208 | 0.008–0.033 | 0.0328 | 0.019–0.047 |
| Bagmati | 801 | 0.0041 | -0.001–0.009 | 0.0203 | 0.009–0.031 | 0.0049 | -0.001–0.01 | 0.0148 | 0.004–0.025 |
| Gandaki | 717 | 0.0067 | 0–0.014 | 0.0309 | 0.017–0.045 | 0.0067 | 0–0.013 | 0.0148 | 0.005–0.024 |
| Lumbini | 194 | 0.0098 | -0.004–0.024 | 0.0699 | 0.033–0.106 | 0.0098 | -0.004–0.024 | 0.0419 | 0.013–0.07 |
| Karnali | 892 | 0.0192 | 0.009–0.03 | 0.0411 | 0.026–0.056 | 0.0230 | 0.011–0.035 | 0.0329 | 0.02–0.046 |
| Sudurpaschim | 360 | 0.0188 | 0.003–0.034 | 0.0226 | 0.005–0.04 | 0.0216 | 0.005–0.038 | 0.0243 | 0.008–0.041 |
| **Sex of household head** | | | | | | | | | |
| Male | 3113 | 0.0144 | 0.01–0.019 | 0.0376 | 0.03–0.045 | 0.0165 | 0.011–0.022 | 0.0253 | 0.019–0.031 |
| Female | 1207 | 0.0091 | 0.003–0.015 | 0.0223 | 0.013–0.031 | 0.0124 | 0.005–0.019 | 0.0187 | 0.011–0.027 |
| **Education of household head** | | | | | | | | | |
| Less than primary | 1735 | 0.0196 | 0.012–0.027 | 0.0369 | 0.027–0.047 | 0.0226 | 0.015–0.03 | 0.0325 | 0.023–0.042 |
| Primary Education | 1362 | 0.0155 | 0.008–0.023 | 0.0370 | 0.026–0.048 | 0.0165 | 0.009–0.024 | 0.0246 | 0.016–0.034 |
| Secondary Education | 920 | 0.0009 | -0.001–0.003 | 0.0270 | 0.015–0.039 | 0.0050 | 0–0.01 | 0.0109 | 0.003–0.019 |
| Higher Education | 304 | 0.0000 | – | 0.0159 | 0–0.032 | 0.0000 | – | 0.0047 | -0.005–0.014 |
| **Consumption quintile** | | | | | | | | | |
| Lowest (First) | 864 | 0.0564 | 0.039–0.074 | 0.0000 | – | 0.0684 | 0.049–0.088 | 0.0349 | 0.022–0.048 |
| Second | 864 | 0.0066 | 0.001–0.013 | 0.0810 | 0.061–0.101 | 0.0066 | 0.001–0.013 | 0.0713 | 0.053–0.09 |
| Middle | 864 | 0.0017 | -0.002–0.005 | 0.0764 | 0.057–0.096 | 0.0017 | -0.002–0.005 | 0.0091 | 0.001–0.017 |
| Fourth | 865 | 0.0000 | – | 0.0072 | 0.001–0.013 | 0.0000 | – | 0.0014 | -0.001–0.004 |
| Highest (Fifth) | 863 | 0.0000 | – | 0.0022 | -0.001–0.005 | 0.0000 | – | 0.0007 | -0.001–0.002 |

Table 6.  Impoverishment impactof healthcare OOP (further below the poverty lines) across background characteristics (N = 4320).

| Covariates | Freq | Absolute poverty lines (international poverty lines) | | | | Relative poverty lines (% consumption of median population) | | | |
|---|---|---|---|---|---|---|---|---|---|
| | | $1.90 per capita per day | | $3.20 per capita per day | | At 50% | | At 60% | |
| | | Value | 95% CI | Value | 95% CI | Value | 95% CI | Value | 95% CI |
| National | 4320 | 0.0038 | 0.0032–0.0044 | 0.0125 | 0.0114–0.0137 | 0.0039 | 0.0033–0.0045 | 0.00688 | 0.0061–0.0077 |
| **Household occupations** | | | | | | | | | |
| Managers and professionals | 195 | 0.0002 | -0.0001–0.0005 | 0.0012 | 0.0001–0.0023 | 0.0002 | -0.0001–0.0005 | 0.00015 | -0.0001–0.0004 |
| Clerical and support workers | 251 | 0.0004 | -0.0003–0.0011 | 0.0041 | 0.0002–0.008 | 0.0004 | -0.0004–0.0013 | 0.00227 | -0.0006–0.0051 |
| Agriculture | 2050 | 0.0040 | 0.0031–0.0049 | 0.0144 | 0.0125–0.0163 | 0.0041 | 0.0032–0.005 | 0.00704 | 0.0059–0.0082 |
| Plant operators and craft workers | 666 | 0.0036 | 0.002–0.0053 | 0.0123 | 0.0094–0.0153 | 0.0038 | 0.0021–0.0055 | 0.00769 | 0.0052–0.0102 |
| Elementary occupations | 881 | 0.0057 | 0.0042–0.0072 | 0.0155 | 0.0129–0.0182 | 0.0059 | 0.0043–0.0074 | 0.00976 | 0.0079–0.0116 |
| Unemployed | 277 | 0.0025 | 0.0008–0.0042 | 0.0055 | 0.0033–0.0077 | 0.0025 | 0.0008–0.0042 | 0.00374 | 0.0015–0.006 |
| Employment status | | | | | | | | | |
| Employed | 4043 | 0.0039 | 0.0032–0.0045 | 0.0130 | 0.0118–0.0143 | 0.0040 | 0.0034–0.0047 | 0.00709 | 0.0062–0.0079 |
| Unemployed | 277 | 0.0025 | 0.0008–0.0042 | 0.0055 | 0.0033–0.0077 | 0.0025 | 0.0008–0.0042 | 0.00374 | 0.0015–0.006 |
| **Sector of occupation** | | | | | | | | | |
| Salaried in non-agriculture | 489 | 0.0015 | 0.0003–0.0027 | 0.0055 | 0.0033–0.0076 | 0.0016 | 0.0003–0.0028 | 0.00361 | 0.0014–0.0058 |
| Salaried in agriculture | 100 | 0.0093 | 0.0037–0.0148 | 0.0250 | 0.0123–0.0376 | 0.0092 | 0.0038–0.0147 | 0.01518 | 0.0082–0.0222 |
| elf-employed in non-agriculture | 470 | 0.0013 | 0.0005–0.0021 | 0.0056 | 0.0031–0.008 | 0.0014 | 0.0005–0.0023 | 0.00321 | 0.0014–0.0051 |
| Self-employed. In agriculture | 3261 | 0.0045 | 0.0037–0.0052 | 0.0148 | 0.0133–0.0164 | 0.0046 | 0.0038–0.0054 | 0.00793 | 0.0069–0.0089 |
| **Area of residence** | | | | | | | | | |
| Urban | 1439 | 0.0015 | 0.001–0.002 | 0.0066 | 0.0056–0.0076 | 0.0016 | 0.001–0.0021 | 0.00293 | 0.0023–0.0036 |
| Rural | 2881 | 0.0049 | 0.0041–0.0058 | 0.0155 | 0.0138–0.0172 | 0.0051 | 0.0042–0.006 | 0.00885 | 0.0077–0.01 |
| **Province** | | | | | | | | | |
| Province one | 693 | 0.0018 | 0.0009–0.0026 | 0.0086 | 0.0062–0.0109 | 0.0019 | 0.001–0.0028 | 0.00394 | 0.0024–0.0055 |
| Madhesh | 662 | 0.0083 | 0.0063–0.0104 | 0.0202 | 0.0168–0.0236 | 0.0085 | 0.0064–0.0106 | 0.01283 | 0.0104–0.0153 |
| Bagmati | 801 | 0.0026 | 0.0014–0.0037 | 0.0094 | 0.0062–0.0125 | 0.0026 | 0.0014–0.0037 | 0.00499 | 0.0034–0.0066 |
| Gandaki | 717 | 0.0020 | 0.0008–0.0032 | 0.0091 | 0.0068–0.0114 | 0.0020 | 0.0008–0.0033 | 0.00350 | 0.0021–0.0049 |
| Lumbini | 194 | 0.0015 | 0.0004–0.0027 | 0.0162 | 0.0095–0.023 | 0.0016 | 0.0003–0.0029 | 0.00688 | 0.0035–0.0102 |
| Karnali | 892 | 0.0047 | 0.003–0.0063 | 0.0155 | 0.013–0.0181 | 0.0049 | 0.0032–0.0066 | 0.00900 | 0.0068–0.0112 |
| Sudurpashchim | 360 | 0.0047 | 0.0025–0.007 | 0.0106 | 0.0069–0.0143 | 0.0049 | 0.0025–0.0072 | 0.00726 | 0.0041–0.0104 |
| **Sex of household head** | | | | | | | | | |
| Male | 3113 | 0.0040 | 0.0032–0.0047 | 0.0131 | 0.0117–0.0146 | 0.0041 | 0.0034–0.0049 | 0.00728 | 0.0063–0.0083 |
| Female | 1207 | 0.0033 | 0.0024–0.0043 | 0.0110 | 0.0091–0.0128 | 0.0034 | 0.0024–0.0044 | 0.00586 | 0.0046–0.0071 |
| **Education of household head** | | | | | | | | | |
| Less than primary | 1735 | 0.0062 | 0.005–0.0074 | 0.0170 | 0.0149–0.0192 | 0.0064 | 0.0052–0.0077 | 0.01048 | 0.0089–0.012 |
| Primary Education | 1362 | 0.0029 | 0.002–0.0037 | 0.0120 | 0.0101–0.0139 | 0.0030 | 0.0022–0.0039 | 0.00647 | 0.005–0.0079 |
| Secondary Education | 920 | 0.0016 | 0.0005–0.0026 | 0.0074 | 0.005–0.0098 | 0.0016 | 0.0005–0.0026 | 0.00256 | 0.0015–0.0036 |
| Higher Education | 304 | 0.0008 | 0.0001–0.0014 | 0.0047 | 0.003–0.0064 | 0.0008 | 0.0001–0.0014 | 0.00127 | 0.0004–0.0021 |
| **Consumption quintile** | | | | | | | | | |
| Lowest (First) | 864 | 0.0182 | 0.0156–0.0208 | 0.0207 | 0.0186–0.0227 | 0.0187 | 0.016–0.0214 | 0.02830 | 0.0255–0.0311 |
| Second | 864 | 0.0007 | -0.0001–0.0016 | 0.0322 | 0.0286–0.0358 | 0.0008 | -0.0001–0.0017 | 0.00496 | 0.0029–0.007 |
| Middle | 864 | 0.0001 | -0.0001–0.0002 | 0.0081 | 0.0047–0.0114 | 0.0001 | -0.0001–0.0002 | 0.00093 | 0–0.0019 |
| Fourth | 865 | 0.0000 | - | 0.0013 | -0.0001–0.0026 | 0.0000 | - | 0.00014 | -0.0001–0.0004 |
| Highest (Fifth) | 863 | 0.0000 | - | 0.0004 | -0.0002–0.001 | 0.0000 | - | 0.00008 | -0.0001–0.0002 |

support worker', whereas the percentage was highest for 'Agriculture' and 'Elementary occupations' at both absolute and relative poverty lines.

Disproportionately greater share of the population from rural areas were pushed further down the poverty compared to the rural area across all the definitions of poverty. Similarly, a larger share of the population from Madhesh, Karnali and Sudurpaschim provinces were disproportionately pushed further down the poverty line compared to other provinces. There was a slight difference in terms of sex of household head, more male headed households were further pushed down the poverty compared to female headed households.

Across the four education categories, the proportion of households pushed further down the poverty was least when the head of household had higher education. Proportion of households pushed further down the poverty increased gradually from least to highest when the household head's education changed from higher education to less than primary education. Similarly, the share of households pushed further down the poverty increased monotonically from highest consumption quintile to lowest consumption quintile. These observations were consistent with all the definitions of poverty.

Fig 2 shows concentration index across occupation, geography and socio-demographic characteristics. A negative (positive) value of concentration index indicates that the burden

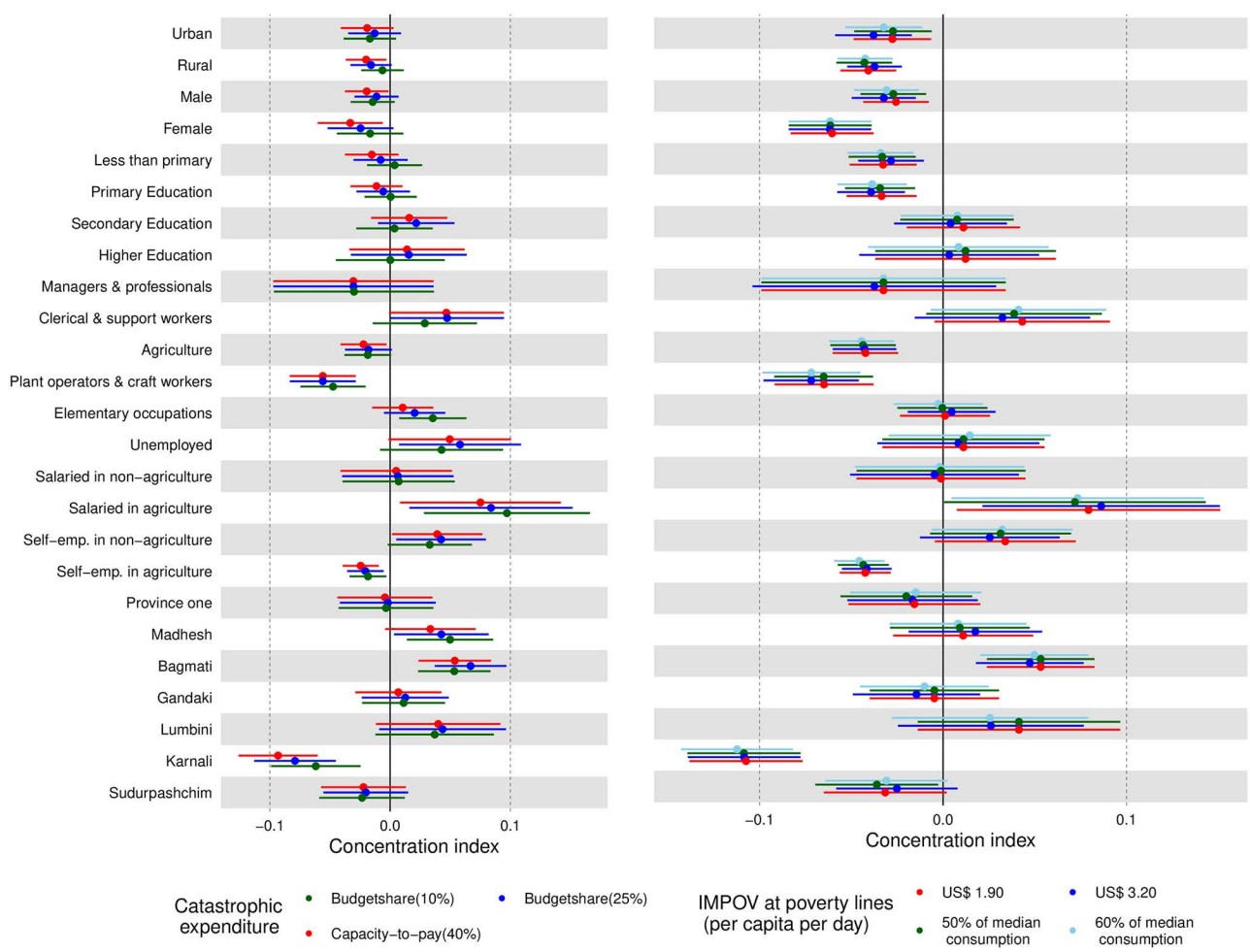

**Fig 2. Concentration index for financial protection indicators across socioeconomic, geography and occupational characteristics.**

of financial hardship falls disproportionally among the poor (rich). Concentration index for CHE using the CTP approach in rural areas was significantly negative. Concentration index for impoverishment impact of healthcare OOP payments at all poverty lines followed the similar pattern in both rural and urban areas. This means that people belonging to lower socioeconomic status were disproportionately becoming poorer due to healthcare OOP payments.

Across the occupation, we found a significantly negative concentration index for both CHE and impoverishment impact for three occupation groups: agriculture, 'plant operators and craft workers', and 'self-employed in the agriculture sector'. This indicates that the burden of CHE and impoverishment impact falls among the relatively poorer segment of the population in these sub-groups. Over the provinces, all the values of the concentration indices were positive for Bagmati province. This indicates that both CHE and impoverishment impacts were disproportionately concentrated among the richer segment of the population. A persistently high CHE even at 25% threshold (Table 4) and a nominal percentage of impoverishment impact (Table 5) together indicate that CHE, when occurs, is mostly concentrated among the richer segment in the Bagmati Province. In the case of Karnali province, concentration indices were negative, indicating that relatively poor households bore a disproportionate burden of CHE and impoverishing healthcare OOP.

## Discussion

This study is the first of its kind in south-east Asia to analyse disaggregated financial protection statistics across occupations and geography. We found CHE of 10.1% and 2.3% at the thresholds of 10% and 25%, respectively using the budget share approach and 4.3% at the threshold of 40% using the CTP approach. Similarly, impoverishment impact was 1.3% and 3.3% at the absolute poverty lines of US$1.90 and $3.20 a day, respectively, and 1.5% and 2.4% at relative poverty lines of 50% and 60% of median consumption. Regarding the share of population (those already below the poverty line) that further pushed into poverty due to healthcare OOP expenditure, an estimated 0.38% was pushed further below the extreme poverty line of PPP US $1.90 per capita per day. At US $ 3.2 per capita per day, an estimated 1.25% of the population was pushed further down the poverty line. Using relative poverty lines of 50% and 60% of median consumption estimated 0.39% and 0.68% of the population, respectively, was further pushed down the poverty. These estimates are comparable to those presented in the global and regional monitoring reports and bulletins published by WHO [2, 30, 46]. Likewise, the consumption expenditure and healthcare OOP are comparable to those presented in the national report published by CBS [40]. The disaggregation of results across occupations could be helpful for policymakers in Nepal in the following ways. First, it shows different levels of vulnerability faced by different occupation groups, and thereby identifies the most vulnerable groups that need government intervention. Second, it provides figures to monitor the financial protection of different occupational groups and assess whether public policies are reducing the inequitable distribution of social security benefits. Third, it provides a rationale for nationwide risk pooling across occupation groups since the current pooling arrangements through National Health Insurance and Social Security Fund are more focused on the informal sector and formal private sector, respectively.

CHE was the lowest (6% at the 10% threshold) for households belonging to 'managers and professionals'. Again, the lowest percentage of the households belonging to this group were pushed below (or further pushed down into poverty) the absolute and relative poverty lines. Our findings are in line with a similar study from South Korea that showed households with employment were less likely to incur CHE (23). Possible explanations behind these findings are: this group of occupations, owing to its formal nature, has a regular income source to

finance uncertain healthcare OOP. As per the report by CBS and ILO these households have the highest salary [47]. Government employees have access to subsidized care at civil servant hospitals [48]. In addition, many private organizations also offer indemnity-based insurance for healthcare to their employees. These households occupy top positions in the consumption quintiles with the most educated household head (Table 2), in which we found relatively well-protected families from financial hardship (Table 4).

Among households with agricultural workers, and 'plant operators and craft workers', both CHE (5.2% and 4.8%, respectively using CTP approach) and impoverishment impact (1.3% and 1.7% at absolute poverty lines of US $1.9 a day) were relatively high. Highest percentage of households belonging to 'elementary occupations' were further pushed down into the poverty, followed by those belonging to agriculture. A large-scale study conducted by Pyakurel et al. in Eastern Nepal also reported a high level of healthcare OOP and CHE among industrial workers [24]. A relatively high CHE and impoverishment due to OOP in this sub-group require attention for several reasons: it is the largest group in terms of occupation (63%), the majority of which fall in the second to fourth consumption quintile and live in rural areas (Table 2). Similarly, the coverage of health insurance schemes is very low in both rural areas and relatively lower consumption quintiles [49]. Likewise, the healthcare OOP share is also among the highest for this group (Table 3). Similarly, this group of households is one of the lowest income groups in comparison to the other occupation counterparts [47] indicating less possibility of mobilizing savings and borrowings to finance OOP. These factors indicate that healthcare OOP put these households at increasing pressure of compromising the use of other goods and services, which makes them prone to financial hardship. Besides this, these households are disproportionately at higher risk due to poor working conditions and surrounding environment. A study conducted by Bhattarai et al. showed that prevalence of work-related injuries and illness was 69% over 12 months study period among agricultural workers in rural areas of Nepal [50]. Other studies also report comparatively higher exposure to cardiovascular risk factors and accidental injuries among industrial workers in Nepal [15, 51].

Households with 'elementary occupations', though less likely to incur CHE (4% at 10% threshold), were more likely to be pushed (and further pushed down into poverty) below the poverty line (1.7% at the absolute poverty line of US $1.90 a day). These members work in the informal sector that does not require special technical skills and perform jobs like selling goods in the street and public places, providing street services, and taking care of homes that need maintenance. These occupations formed one-fifth of the total labour market in Nepal [47]. Bonnet et al. reported that people in these occupations work in a compromised environment [9]. Such working conditions put these individuals and households at a relatively higher risk of being ill, resulting in poor health and increased healthcare needs. In this connection, a study reported that this group of households was in the third rank to occupy the majority of occupational diseases [52]. On the other hand, these households are characterized by a lower and risky income, as reported by a few studies in developing countries [9]. Likewise, our findings also report that these households lie in the lowest first two consumption quintiles. These pieces of evidence indicate that lower income coupled with income uncertainty and lower welfare in terms of consumption keep these households under increasing pressure to meet the regular household requirements. On top of this, a vast majority of them live in rural areas, and those living in urban areas are living in unhealthy living conditions [15]. These circumstances made healthcare less affordable to them, so the workers belonging to informal sector may have forgone care due to financial barriers. This is one of the reasons for a relatively lower level of CHE in this sub-group. However, these households faced a disproportionately greater burden of the impoverishment impact of healthcare OOP. The quintile distribution of elementary occupation reveals that more than half (55%) of the households with elementary occupations

lie in the bottom two consumption quintiles (Table 2). This indicates that per capita consumption for the majority of such households is marginally above the poverty line before healthcare OOP payments. And, any healthcare OOP (not more than 10% of the total consumption which does not causes CHE) is likely to lower the consumption per capita net of healthcare OOP less than poverty lines thereby impoverishing them without incurring any CHE. This pattern is commonly observed in the poor segment of the population [46]. As a response to this situation, the government of Nepal has launched many social protection schemes focusing particularly towards poor and vulnerable population groups belonging to the informal sector. A few examples are: the implementation of national health insurance, impoverished citizen treatment scheme, and free basic health services. National health insurance is fully subsidised for identified poor households. However, the result of the poor household identification survey is available for only 26 districts. This has put poor households from remaining districts in a disadvantaged position as they are not being able to benefit from this policy decision. For those households who are enrolled (either through contribution or subsidy) due to inequitable distribution of healthcare providers across provinces, people have to travel to a nearby city or Kathmandu where the cost of treatment could be higher than that could have been in the nearby area. Public facilities in remote areas are understaffed and frequently face a stock out of medicines. Due to this, the people in such areas either have to rely on private clinics and tertiary hospitals in the cities. Due to all these reasons, people from remote and rural areas, who are also from the informal sector, get disproportionately disadvantaged. To tackle this, poor households from all the districts should be identified so that they can benefit from public subsidies, healthcare should be more equitably distributed and human resources and logistics should be ensured in the rural and remote areas. The three tiers of government should work together so that the households eligible for health insurance subsidies could be identified and basic health services should be insured at the local level so that people do not have to pay a large sum of money to private and tertiary hospitals even to cure minor ailments.

Across the area of residence, using budget share approach, we observed no clear difference in CHE, but using CTP approach, 5% and 3% of households faced catastrophic health expenditure in the rural and urban areas, respectively. The CTP approach deducts food consumption expenditure from the total consumption when measuring CHE. NLSS-III shows that food share of total consumption is higher in rural areas [53]. The CTP approach is therefore a pro-poor measure [39] and provides a better measure of CHE and hence explains observed higher level of CHE in rural areas when budget share approach does not provide any evidence of systematic differences. In the same line, impoverishing health expenditure and poverty gap (households below poverty line that were further pushed down into poverty) were quite high in rural areas across all the definitions of poverty lines. In both areas, relatively poor households disproportionately borne the burden of CHE and impoverishment impact. The rural population is relatively poorer, as reported by the series of NLSS [53, 54]. Similarly, the population in rural areas have less access to healthcare in comparison to the urban population [55, 56]. Likewise, healthcare delivery in rural areas is mostly dominated by public facilities with primary healthcare services, and higher-level care is not available [57]. This greatly decreases the utilization of healthcare in the rural population, and therefore, households in this area are less likely to incur CHE. However, when they do, in most cases, healthcare OOP is impoverishing due to a higher level of poverty and a lower level of consumption expenditure in rural households [53]. A series of WHO reports on financial protection that covers the experience from LMICs also showed that CHE is higher in urban areas and lower in rural areas [2, 46].

Across the provinces, CHE was lowest in Province 1 and Sudurpaschim, and highest in Madhesh and Lumbini. The Department of Health Services (DoHS) reported the highest

hospital density in Province 1 indicating availability of tertiary care in these areas [58]. Likewise, Bhusal and Sapkota reported a relatively higher coverage of health insurance schemes in this province [49]. Similarly, this province covers a relatively lower share of poor households [59]. This explains a relatively lower level of CHE in Province 1. However, the lower level of CHE in Sudurpaschim province requires further studies. This province accommodates relatively poorer households in Nepal [59] and access to tertiary care is limited to just a few private hospitals [58]. These reasons explain the lower level of CHE in this province. We found a relatively higher level of impoverishing healthcare OOP in Karnali, Madhesh and Lumbini Province. Similarly, higher level of poverty gap was found in Karnali, Madhesh and Sudurpaschim province. In these provinces also a major share of the population belongs to the poor category [59], which makes the residents more likely to incur impoverishing healthcare OOP. The federal structure of the country provides opportunity to provincial and local governments to develop community-based targeting of poor households in absence of poor-households identification surveys from the federal government. This will help poor households from the rest of the districts benefit from the health insurance subsidy. To improve cross-subsidization health insurance schemes should be made mandatory for both formal and informal sectors of workers. Local governments are in good position to enforce mandatory enrolment by linking the membership with other social services and benefits provided through the local units (wards and municipalities).

## Strength of this study

This study is the first of its kind to analyse the situation of financial protection across occupations and geography based on a nationally representative cross-sectional survey. AHS employed a standard consumption measurement tool based on COICOP, which ensures consistency in the use of definitions and international comparison of the findings. Similarly, we used occupational classification that corresponds to ISCO, making the results comparable to the studies from other countries adopting similar occupational classifications. Similarly, the study also used widely accepted methods to measure financial protection, covering nearly all the definitions. The produced statistics can be useful to planners and policymakers engaged in designing and monitoring health financing reform, including risk-pooling arrangements to make progress towards UHC.

## Limitation of this study

This study is not without limitations. First, occupation is an individual-level variable. We used it to categorize the households based on the major occupation of the household. The classification, however, has produced relevant findings for the policymakers and are consistent with the theory and literature as discussed in the introduction section. Second, though our findings are based on a survey from 2014–15, this is the most recent survey from Nepal in terms of covering information on occupation and consumption expenditure. After the survey, there are no major changes in the labour policy and overall economic and development paradigms in Nepal. So, the composition of the occupational groups is still relevant. Third, we found a low level of impoverishment impact across some categories of occupation that have a low frequency (less than 10%). More focused studies in these groups are likely to reveal the exact situation. Since we aimed to measure financial protection indicators in this paper, detail discussion of the other dimensions of UHC, such as the supply side of the health system, was outside the scope of the current work. Fourth, we could not provide an account of foregone care, an important and complementary analysis to the financial protection measures, due to unavailability of such data in the AHS.

## Conclusion

Despite the global and national efforts to reduce the financial burden through various health financing reforms, we observed that financial protection is unequal between and within occupation groups and geography in Nepal. The formal occupations have a lower share of total occupations in Nepal and are protected to a greater degree from the financial hardship resulting from healthcare OOP payments than the informal occupations such as agricultural workers and 'plant operators and craft workers' that occupy two thirds of all the occupations in Nepal. Additionally, to make the things worse, the problem is also concentrated in poorer segment of informal occupations. We also observed a high CHE in relatively richer provinces and rural areas when using pro-poor measurement approaches (CTP). In addition, the impoverishment impact and poverty gap were high in relatively poorer provinces and rural areas. These findings suggest a need for policies and strategies that generate incentives for effective uptake of the informal occupations in the risk-pooling arrangements, and address the occupational, provincial and urban/rural differences in financial protection by harnessing the potential of federal structure and local units.

## Supporting information

**S1 File.**
(DOCX)

## Acknowledgments

The authors would like to acknowledge the CBS and Annual Household Survey (AHS) for their permission to access and use the dataset for this study.

## Author contributions

**Conceptualization:** Vishnu Prasad Sapkota, Umesh Prasad Bhusal.

**Data curation:** Vishnu Prasad Sapkota, Govinda Prasad Adhikari.

**Formal analysis:** Vishnu Prasad Sapkota, Umesh Prasad Bhusal.

**Investigation:** Vishnu Prasad Sapkota, Govinda Prasad Adhikari.

**Methodology:** Vishnu Prasad Sapkota, Umesh Prasad Bhusal, Govinda Prasad Adhikari.

**Project administration:** Vishnu Prasad Sapkota, Umesh Prasad Bhusal.

**Resources:** Vishnu Prasad Sapkota, Govinda Prasad Adhikari.

**Software:** Vishnu Prasad Sapkota.

**Validation:** Vishnu Prasad Sapkota, Umesh Prasad Bhusal.

**Visualization:** Vishnu Prasad Sapkota.

**Writing – original draft:** Vishnu Prasad Sapkota, Umesh Prasad Bhusal, Govinda Prasad Adhikari.

**Writing – review & editing:** Vishnu Prasad Sapkota, Umesh Prasad Bhusal, Govinda Prasad Adhikari.

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
