## [Decision Letter · Decision Letter 0]

11 Sep 2022

PONE-D-22-17102Occupational and geographical differentials in financial protection against healthcare out-of-pocket payments in Nepal:  evidence  for universal health coveragePLOS ONE

Dear Dr. Sapkota,

Thank you for submitting your manuscript to PLOS ONE. After careful consideration, we feel that it has merit but does not fully meet PLOS ONE’s publication criteria as it currently stands. Therefore, we invite you to submit a revised version of the manuscript that addresses the points raised during the review process.

We look forward to receiving your revised manuscript.

Kind regards,

Kuo-Cherh Huang

Academic Editor

PLOS ONE

Journal Requirements:

Additional Editor Comments:

Dear Mr. Sapkota,

We appreciate your submission to PLoS ONE. Although your paper may be of interest to health policymakers in Nepal, both reviewers have provided a variety of important concerns and helpful suggestions. In particular, I concur with the following comment of Reviewer 2 after my own review of the paper, “it is a bit overwhelming for readers to navigate through the paper and the level of details especially in the results section. The elaborate description of findings at times is rather dry.” Please respond carefully and thoroughly to each comment of the reviewers.

Kuo-Cherh Huang

Reviewers' comments:

Reviewer's Responses to Questions

**Comments to the Author**

1. Is the manuscript technically sound, and do the data support the conclusions?

Reviewer #1: Yes

Reviewer #2: Yes

2. Has the statistical analysis been performed appropriately and rigorously?

Reviewer #1: Yes

Reviewer #2: Yes

3. Have the authors made all data underlying the findings in their manuscript fully available?

Reviewer #1: Yes

Reviewer #2: Yes

4. Is the manuscript presented in an intelligible fashion and written in standard English?

Reviewer #1: Yes

Reviewer #2: No

5. Review Comments to the Author

Reviewer #1: As I stated in my comments, you need to include more references in your manuscript because there are studies out there, looking at causal relations. Also, including more years of data would be very important for your analysis.

Reviewer #2: The is a descriptive study that aims to understand the differences in financial protection among different population group (by occupation type, geography etc) in Nepal using the annual household survey data from 2014-2015. The authors employ several measures of financial protection including catastrophic health expenditure (CHE), impoverishment due to OOP health expenditure, as well as inequality in financial risk protection using concentration index.

The paper provides a detailed descriptive summary of findings across all the metrics. However, my general comment is that it is a bit overwhelming for readers to navigate through the paper and the level of details especially in the results section. The elaborate description of findings at times is rather dry. There is a lot of room to condense the results section, by focusing only on the key findings in the description and referring to the tables for the others, for example. What is more useful is for readers to understand the context and the interpretation of the findings which can be strengthened.

The paper would benefit from language editing as there are long convoluted sentences throughout the paper, which can be simplified for clarity.

Specific comments

• Introduction: Please solidify the rationale for your study. It’s stated that there is no disaggregated information (which is an important rationale) but can substantiate with why the disaggregated information would be useful and how it may contribute to the literature or in policy decision processes. Some of these points are listed as bullet points (in page 5) but can be developed and simplified further for clarity.

• Methods: Independent variables for what? It would be clearer to first describe what you intend to do and then describe the independent (and dependent) variables rather than starting by calling out independent variables.

• Measures of consumptions- not well defined/explained. Elaborate for clarity. What are the different approaches supposed to mean? Are you testing or validating the results with different approaches? If so, you may want to highlight any differences or deviations in findings between the approaches? Also how do you console your findings if you find differences? In the discussion section there is some elaboration on why the differences were observed and the justification (which is good). You may want to state the intention of using different measures in the beginning to help readers follow through the manuscript. For example, what do the different thresholds mean? Why use multiple thresholds/measures and what they mean?

• Results: The tables are a bit long, but they provide a comprehensive descriptive summary (Tables 3, 4). In table 3, are the estimates of consumption per capita or per household. Please clarify.

• The results section is too elaborate and dry. It could be condensed by focusing only on the key findings and referring to the tables for the others. What would be more useful for readers is to understand the context and the interpretation of the findings which to some extent is covered in the discussion section. A cohesive story/interpretation from the descriptive findings is lacking. I would suggest building that which would enhance the utility of these kinds of studies.

• Why would you expect differences by sex of household head, education status and all of what you refer as independent variables? Elaborating this would help contextualize the results.

• Interesting finding that both CHE and impoverishment impact were higher among the richer population in Bagmati province, compared to other. Why would that be the case?

• If the access to health care is limited, then the CHE is likely going to be small. But how will this lead to impoverishment? Not clear, additional explanation would be helpful (page 20)

• Discussion: No systematic differences in catastrophic health expend across residence by using one method (budget share approach) but some difference using another (CTP). Why would that be the case? How do you explain the differences?

• Conclusion: There is too much of information in the results/discussion section, prioritization of some key takeaways would be useful to include in the conclusion section rather than restating some of the result figures.

6. PLOS authors have the option to publish the peer review history of their article (what does this mean?). If published, this will include your full peer review and any attached files.

Reviewer #1: No

Reviewer #2: No

---

## [Author Response · Author response to Decision Letter 0]

7 Oct 2022

Response to Editor’s remarks

Response: We have now changed the manuscript formatting that meets PLOS ONE style.

2. In your Data Availability statement, you have not specified where the minimal data set underlying the results described in your manuscript can be found.

Response: We have attached the raw data as supporting information file-I, that covers data, survey report, questionnaire and field manual.

Response: This paper is based on a social survey conducted by Central Bureau of Statistics, Nepal, and it is currently publicly available for further analysis. We have now included ethical considerations in the method section as follows.

…. This study is based on a publicly available dataset. The permission to access and use this dataset was obtained from Central Bureau of Statistics (CBS), Government of Nepal. The protocol for the survey was approved by the Central Bureau of Statistics (CBS) based on the Statistical Act (1958) in September 2018. This Act enables CBS to implement surveys as per the Government of Nepal’s ethics protocol without involving an institutional review board (IRB) approval. So, all procedures were performed in accordance with relevant guidelines. During the data collection, verbal consent was obtained from each respondent after a thorough introduction of the survey. All respondents were briefed about the voluntary nature of participation. Participants were assured anonymity and confidentiality of shared information.

Besides this, considering the editor’s emphasis on the second reviewer’s remark – prepare a cohesive story from the result section – we given attention to this point and revised the respective sections accordingly.

RESPONSE TO THE COMMENTS FROM REVIEWER 1

The study investigates financial protection using two approaches in Nepal. They disaggregate their analysis based on occupations and geography.

1. Cross-sectional AHS 2014-15: Authors use one-year cross-section data to do their analysis. Using one-year data would not tell us much about what is going on with respect to financial protection. If possible, I would suggest adding more past data to see the country's CHE trend over the years.

Response: Many thanks for this important suggestion. Our aim was to examine the financial protection indicators, both levels and distribution for the most recent annual household survey (AHS) that measures both consumption and occupational details, thus disaggregation at occupation and geography was possible. There are additional reasons why this round of AHS was used. In terms of timeframe this corroborates with the baseline year for Sustainable Development Goals (SDGs). So, the findings reported in this study could be used to track the changes to be reported by subsequent surveys. Post 2015, the government of Nepal has introduced various health financing reforms, notably National Health Insurance (launched in 2016) in an attempt to curb down the financing barriers to healthcare. The scheme, now launched in all 77 districts of Nepal, is fully subsidized for the poor and vulnerable households. So, in the subsequent work we will try to incorporate the findings form the surveys from the years conducted after 2016. This will contribute towards monitoring of the financial risk protection, one of the key dimensions of universal health coverage.

2. The authors suggest designing risk-pooling arrangements to make progress towards UHC. Will this solve all the problems people face in Nepal, especially the most disadvantaged groups incurring CHE? How about the supply side of the health care system? If there are not enough provisions for these services, how come, UHC will help solve CHE? I would recommend talking about the supply side of the health care system in Nepal.

Response: Many thanks for raising this important concern. We have revised the background section focusing on health financing instruments. Still, we have focused more on the “risk-pooling arrangements” together with “health financing arrangements” since the former is more closely associated with financial protection because the way revenue for health are generated and pooled affects who bear the burden of healthcare payments and who utilizes the healthcare. So appropriately designed health financing arrangements help make sure people are paying for healthcare based on their ability to pay (socio-economic position in the society) and using the healthcare based on their need (equity). So, health financing arrangements like public health insurance are designed and launched to provide financial protection to vulnerable populations against health shocks (such as CHE).

Here, our aim was to analyze the levels and distribution of financial risk protection, a demand side concept that measures the opportunity cost of healthcare OOP payments in terms of reduced household consumption of other goods and services. That is why we did not talk much about the supply side of the health system. The framework of health financing and UHC inherently captures ‘service coverage’ or ‘benefit package’ as key functions or dimensions. The revenue generated and pooled by health financing institutions (example: health insurance agency) is used to purchase healthcare on behalf of the population. The financial power of such institutions can ensure the availability and quality of healthcare.

Since the aim of the paper, as stated earlier, was to talk about one of the dimensions of UHC (financial protection) we were unable to include the aspects of the supply-side of healthcare. However, this should not imply that only the pooling function guarantees UHC. We have reviewed the Manuscript and tried to capture the aspects of supply-side as well, wherever applicable. In addition, we have added this line in the limitation section:

“Since we aimed to measure financial protection indicators in this paper, detailed discussion of the other dimensions of UHC, such as the supply side of the health system, was outside the scope of the current work.”

3. Introduction: "In the last few decades, many low-and middle-income countries (LMIC) have been reforming their health financing arrangements for UHC (4, 5). Countries such as Thailand, Ghana, Tanzania, and Indonesia have successfully introduced the risk-pooling arrangements to the occupation groups of both the formal and informal sector of economy (4-6)."

a. Turkey introduced universal health coverage in 2008 and offered a good example of such countries that implemented UHC successfully. Therefore, I would suggest including the below references in the text as well to mention Turkey's experience:

Tirgil, A., & Ozbugday, F. C. (2020). Does Public Health Insurance Provide Financial Protection Against Out-Of-Pocket Health Payments? Evidence from Turkey. Sosyoekonomi, 28(45), 11-24.

Atun, R., Aydın, S., Chakraborty, S., Sümer, S., Aran, M., Gürol, I., ... & Akdağ, R. (2013). Universal health coverage in Turkey: enhancement of equity. The Lancet, 382(9886), 65-99.

Response: We appreciate your suggestion to expand our literature and references. In the revised manuscript we have included Turkey's example as well. Following text is added in the ‘Introduction’ section:

“Turkey introduced a non-contributory health insurance scheme named Green Card in 1992 (further revised in 2005 to include outpatient services and pharmaceuticals) to provide financial protection to poor households that were previously not covered by the formal health insurance scheme. Health system and financing reforms under Health Transformation Program in Turkey are successful in reducing inequity in health outcomes and making progress towards universal coverage.”

4. "There are two commonly used approaches to measure the extent of financial hardship and financial protection in healthcare: (i) catastrophic health expenditure (CHE) and (ii)impoverishment due to healthcare out-of-pocket (OOP) payments."

Other studies in the literature look at causal mechanisms in terms of catastrophic health expenditures. Therefore, I suggest that the authors provide references to the relevant literature. For example,

Miller, G., Pinto, D. M., & Vera-Hernández, M. (2009). Risk protection, service use, and health outcomes under Colombia's health insurance program for the poor (No. w15456). National Bureau of Economic Research.

Response: We followed the examples from WHO and World Bank who used catastrophic health expenditure (CHE) and impoverishment due to healthcare out-of-pocket (OOP) payments as approaches to measure the extent of financial hardship and financial protection in healthcare (see Global Monitoring Report on Financial Protection in Health series such as this https://www.who.int/publications/i/item/9789240040953. We used the approaches described in these monitoring reports to measure financial protection at different thresholds. Adopting their approach allowed for comparison across countries and years since the level/measure of the financial risk may depend on the methodology and definition used.

5. There is much research concerning CHE using these two approaches authors utilize. I do not see any contribution to the literature regarding the approaches taken. However, the authors state that it is the first study to look at CHE, disaggregated at occupations and regions in South-East Asia, which can be accepted as a contribution.

Response: We agree with the reviewer that we did not introduce any new methodological approaches to measure financial protection in health. Rather, we used standard approaches to measure CHE and impoverishment due to OOPs commonly found in World Bank and WHO reports. However, the novelty of our work lies in the fact that we estimated the financial risk protection indicators at different occupation groups and geography (provinces, rural/urban). This will be particularly important given that Nepal has transitioned to Federal structure since 2017 from the previous unitary government structure and the government has also launched social health protection schemes targeted to different occupation groups (Example: Social Security Schemes launched targeting formal private sector, National Health Insurance launched targeting the informal workforce).

Response to the comments from reviewer 2

Thank you for raising general issues with the paper, we have now tried to address these points including cohesive story around findings and the grammatical aspect. Below are the response to each specific points.

1. Introduction: Please solidify the rationale for your study. It’s stated that there is no disaggregated information (which is an important rationale) but can substantiate with why the disaggregated information would be useful and how it may contribute to the literature or in policy decision processes. Some of these points are listed as bullet points (in page 5) but can be developed and simplified further for clarity.

[Response] We thank the reviewer for important suggestions on potential contributions to the existing literature and relevance to policy makers. We have improved the rationale addressing these points in the following text in the last paragraph of the introduction section of the paper.

…In this context, our study aims to examine the financial protection statistics, both levels and distribution, across occupation and geography using the nationally representative Annual Household Survey (AHS). Currently, such measures are available only at the aggregate/national level. Besides this, the results of this study can be useful in several ways. First, the GoN through social protection schemes targets the population in different occupations. Therefore, this study will provide baseline, disaggregated estimates of financial protection to monitor the performance of existing risk-pooling arrangements. Second, the financial protection of occupations engaged in informal sectors (which occupies 62% share) (8) is particularly unknown. The coverage of these groups in existing social health protection schemes is also poor (19). Having evidence on financial protection can be useful to policy makers to devise strategies, garner budgetary requirements and monitor effectiveness of the policies over time. Third, rural-urban and provincial disaggregation is important in light of the fact that the country's federal structure provides provincial, local planners & policymakers a significant budgetary discretion in devising strategies to improve financial protection. Fourth, occupational classification indicators move together with the size and composition of the economy, which is an important precondition for progress towards UHC (4, 29). Fifth, each occupation group covers workers falling on a range in a socio-economic spectrum; therefore, it is relevant to check, within each occupation group, whether the financial protection statistics are concentrated more among the rich or poor. This piece of evidence can be useful in devising strategies to bridge the rich-poor gap in financial protection within each occupation group, an important aim of UHC….

 Methods: Independent variables for what? It would be clearer to first describe what you intend to do and then describe the independent (and dependent) variables rather than starting by calling out independent variables.

[Response] We consider this a reasonable point and hence improved the flow of the methodology section as advised by the reviewer, first by highlighting our main dependent variable – financial protection, its measurement – and, then listed the independent variables that are suitable given the aim of this paper and data availability.

2. • Measures of consumptions- not well defined/explained. Elaborate for clarity. What are the different approaches supposed to mean? Are you testing or validating the results with different approaches? If so, you may want to highlight any differences or deviations in findings between the approaches? Also how do you console your findings if you find differences? In the discussion section there is some elaboration on why the differences were observed and the justification (which is good). You may want to state the intention of using different measures in the beginning to help readers follow through the manuscript. For example, what do the different thresholds mean? Why use multiple thresholds/measures and what they mean?

[Response] We thank the reviewer for highlighting the important point. Measurement of financial protection, besides healthcare OOP payments, requires a measure of household resources available-to-pay. In the absence of an ideal definition of household resources available-to-pay, there are commonly used proxy measures for it. For these proxy measures, the starting point is a measure of household consumption expenditure. We have now elaborated the components of household consumption expenditure as follows, and added references that provide details about it for interested readers.

….The AHS used the Classification of Individual Consumption by Purpose (COICOP) to classify household consumption expenditures into 12 broad categories. The categories cover food (COICOP-1 and 11), and non-food consumption expenditures (COICOP 2, 12) that includes tobacco and alcohol, clothing, housing, furnishing, transportation, communication, entertainment, health, education, and other expenses. The details about the components of consumption are available in the AHS report (41)….

In the literature of catastrophic health expenditure, there is no ideal approach of measurement. The benefit of using multiple approaches is that one measurement approach captures shortcomings of the other, and with collective interpretation, the author can make a rich analysis of the findings and can be useful to the intended audience. In the methodology section, besides elaborating each measurement approach, we have added the significance of multiple approaches, which will be used in the ‘findings’ and ‘discussion’ sections of the paper. We have added the following text to address this issue.

…The argument behind deducing a certain amount from the total consumption in (ii) and (iii) definition is to capture a better measure of a household’s ability or capacity-to-pay healthcare OOP payments (39). With the (i) definition, the CHE is usually less concentrated among “poor people” (or more concentrated among “rich people”). In order to address this bias at lower tail of income distribution, a pro-poor measure called the CTP approach is used that records a higher incidence of CHE spending among the poor than the budget share approach. A comparative analysis of these two approaches provides a balanced understanding of CHE (40). We used the first two approaches to measure financial protection indicators so that the results will be comparable to the available literature including the global reports on financial protection published by the WHO and the World Bank (8)….

As elaborated in the ‘methods’ section, shortcoming of one measurement approach is fulfilled by the other and hence reporting them together provides a complete picture. Similarly, we have also highlighted differences between the measurement approaches and their usefulness in measuring financial protection. Additionally, the reasons for using multiple thresholds is also elaborated in the paper (in methodology section).

3. • Results: The tables are a bit long, but they provide a comprehensive descriptive summary (Tables 3, 4). In table 3, are the estimates of consumption per capita or per household. Please clarify.

[Response]: Thank you. We agree with the reviewer, the numbers indicate per capita per year and have changed the title accordingly in the revised version.

4. • The results section is too elaborate and dry. It could be condensed by focusing only on the key findings and referring to the tables for the others. What would be more useful for readers is to understand the context and the interpretation of the findings which to some extent is covered in the discussion section. A cohesive story/interpretation from the descriptive findings is lacking. I would suggest building that which would enhance the utility of these kinds of studies.

[Response] A very pertinent remark by the reviewer. The CHE, measured using two approaches, provides a comprehensive understanding of the welfare impact of healthcare OOP payments. Therefore, we have stated the important findings based on both budgetshare and CTP approach. These findings, together with impoverishment impact at different poverty lines, will be useful in the discussion. Based on reviewers’ suggestions, we have made the following changes in the findings section.

1. For table 2 and 3, we have highlighted the important differences that provide merit on its own, and are useful in explaining the observed differences shown in table 4 and 5, leaving the rest of the findings in the table.

2. For table 4 and 5, we have now highlighted the main findings/statistics related to financial protection highlighting different measurement approaches together so that readers can understand the meaning considering the facts highlighted in the methodology section. We have again considered these key findings in the discussion section.

5. • Why would you expect differences by sex of household head, education status and all of what you refer as independent variables? Elaborating this would help contextualize the results.

[Response] Important point! Our aim is to highlight and explain the observed difference in financial protection across occupation and geography. We also got disaggregation across a set of sociodemographic characteristics – household head’s sex, household head’s education and consumption quintile of household. The choice of these variables is based on the review of the latest Labour Force Survey (LFS) 2018/19, which reported that occupation groups differ across gender, education and socioeconomic variables. So, disaggregating both occupation groups and financial protection across sociodemographic characteristics is helpful to explain if such socio-economic variation in occupation categories also explains the differences in financial protection across occupations. We have used these statistics in the discussion section wherever relevant.

6. • Interesting finding that both CHE and impoverishment impact were higher among the richer population in Bagmati province, compared to other. Why would that be the case?

[Response] Yes, the elaboration will be helpful for readers. These statistics reveal important aspects about how CHE, at different thresholds, and impoverishment impact can be used to explain this observed difference. We have added the following text to explain this observation.

…Over the provinces, all the values of the concentration index were positive for Bagmati province. This indicates that both CHE and impoverishment impacts were disproportionately concentrated among the richer segment of the population. This occurs because a persistently high CHE even at 25% threshold (Table 4) and a nominal percentage of impoverishment impact (Table 5) together indicate that CHE, when occurs, is mostly concentrated among the richer segment in the Bagmati Province…

7. • If the access to health care is limited, then the CHE is likely going to be small. But how will this lead to impoverishment? Not clear, additional explanation would be helpful (page 20)

[Response] This is a policy relevant observation in the financial protection literature. In general, those households lying slightly above the poverty line are not poor. It is commonly reported that such households have poor access to healthcare as the majority of them face different barriers to access healthcare. We have added the following text in the discussion to explain it.

…The quintile distribution of elementary occupation reveals that more than half (55%) of the households with elementary occupations lie in the bottom two consumption quintiles (Table 2). This indicates that per capita consumption for the majority of such households is marginally above the poverty line before healthcare OOP payments. And, any healthcare OOP (not more than 10% of the total consumption which does not cause CHE) is likely to lower the consumption per capita net of healthcare OOP less than poverty lines thereby impoverishing them without incurring any CHE. This pattern is commonly observed in the poor segment of the population (47)….

8. • Discussion: No systematic differences in catastrophic health expend across residence by using one method (budget share approach) but some difference using another (CTP). Why would that be the case? How do you explain the differences?

[Response] Thank you for indicating the important point. This finding reveals the usefulness of pro-poor measure of CHE. We have explained this observation by adding the following text in the discussion.

….Across the area of residence, using the budget share approach, we observed no clear difference in CHE, but using the CTP approach, 5% and 3% of households were catastrophic in the rural and urban areas, respectively. The CTP approach deduces food consumption expenditure from the total consumption when measuring CHE. NLSS-III shows that the food share of total consumption is higher in rural areas (54). The CTP approach is, therefore, a pro-poor measure(40) and provides a better measure of CHE and hence explains observed higher level CHE in rural areas when budget share approach does not provide any evidence of systematic differences. In the same line, impoverishing health expenditure was quite high in rural areas across all the definitions of poverty lines. …

9. • Conclusion: There is too much of information in the results/discussion section, prioritization of some key takeaways would be useful to include in the conclusion section rather than restating some of the result figures.

[Response]: Now we have reframed the conclusion to explain the key takeaway from the paper.

….Despite the global and national efforts to reduce the financial protection through various health financing reforms, we observed that financial protection is unequal among and within occupation groups and geography in Nepal. The formal occupations – having lower share of total occupations in Nepal – are protected to a greater degree from the financial hardship resulting from healthcare OOP payments than the informal occupations such as agricultural workers and ‘plant operators and craft workers’ that occupy two thirds of all the occupations. Additionally, to make things worse, the problem is further concentrated in poorer segments of informal occupations. We also observed a high CHE in relatively richer provinces and rural areas when adjusted with pro-poor measurement approaches. On the other hand, the impoverishment impact was found high in relatively poorer provinces and rural areas. These findings suggest a need for policies and strategies that generate incentives for effective uptake of the informal occupations in the risk-pooling arrangements, and also address the provincial and urban/rural differences in financial protection by harnessing the benefits of federal structural and independent local units….

---

## [Decision Letter · Decision Letter 1]

1 Dec 2022

PONE-D-22-17102R1 Occupational and geographical differentials in financial protection against healthcare out-of-pocket payments in Nepal:  evidence  for universal health coverage PLOS ONE

Dear Dr. Sapkota,

Thank you for submitting your manuscript to PLOS ONE. After careful consideration, we feel that it has merit but does not fully meet PLOS ONE’s publication criteria as it currently stands. Therefore, we invite you to submit a revised version of the manuscript that addresses the points raised during the review process.

We look forward to receiving your revised manuscript.

Kind regards,

Kuo-Cherh Huang

Academic Editor

PLOS ONE

Journal Requirements:

Additional Editor Comments :

Dear Mr. Sapkota,

Thank you for submitting your revised manuscript to PLoS ONE. I have carefully read your revised manuscript and responses to the previous round of review comments. I appreciate your efforts. Nonetheless, one reviewer had raised several vital concerns regarding your study, and had provided suggestions to improve your work further, particularly the Discussion section. Basically, I am in agreement with the reviewer’s concerns. In light of the concerns, I invite you to respond to the reviewer's comments and resubmit a revised manuscript. Please respond carefully and thoroughly to each comment of the reviewer. Please explain where you feel you cannot completely agree with the reviewer’s suggestions.

Kuo-Cherh Huang

Reviewers' comments:

Reviewer's Responses to Questions

**Comments to the Author**

1. If the authors have adequately addressed your comments raised in a previous round of review and you feel that this manuscript is now acceptable for publication, you may indicate that here to bypass the “Comments to the Author” section, enter your conflict of interest statement in the “Confidential to Editor” section, and submit your "Accept" recommendation.

Reviewer #1: All comments have been addressed

Reviewer #2: All comments have been addressed

Reviewer #3: (No Response)

2. Is the manuscript technically sound, and do the data support the conclusions?

Reviewer #1: Yes

Reviewer #2: Yes

Reviewer #3: Yes

3. Has the statistical analysis been performed appropriately and rigorously? 

Reviewer #1: Yes

Reviewer #2: Yes

Reviewer #3: Yes

4. Have the authors made all data underlying the findings in their manuscript fully available?

Reviewer #1: Yes

Reviewer #2: Yes

Reviewer #3: Yes

5. Is the manuscript presented in an intelligible fashion and written in standard English?

Reviewer #1: Yes

Reviewer #2: Yes

Reviewer #3: Yes

6. Review Comments to the Author

Reviewer #1: (No Response)

Reviewer #2: While the paper could further be condensed, the authors have satisfactorily addressed the comments.

Reviewer #3: While the paper is well written and well analyzed, it is not really original. These kinds of analysis have been done a lot, and the only thing new that is done here is the disaggregation by occupation. The paper should definitely be published. The question is, should it be published in Plos One? If Plos one values having original papers and research questions, then this is not a good fit.

7. PLOS authors have the option to publish the peer review history of their article (what does this mean?). If published, this will include your full peer review and any attached files.

Reviewer #1: No

Reviewer #2: No

Reviewer #3: No

---

## [Author Response · Author response to Decision Letter 1]

6 Jan 2023

Editor’s comment

Thank you for submitting your revised manuscript to PLoS ONE. I have carefully read your revised manuscript and responses to the previous round of review comments. I appreciate your efforts. Nonetheless, one reviewer had raised several vital concerns regarding your study, and had provided suggestions to improve your work further, particularly the Discussion section. Basically, I am in agreement with the reviewer’s concerns. In light of the concerns, I invite you to respond to the reviewer's comments and resubmit a revised manuscript. Please respond carefully and thoroughly to each comment of the reviewer. Please explain where you feel you cannot completely agree with the reviewer’s suggestions.

Response: Dear Editor, we are grateful to you and the third reviewer for providing us an opportunity to revise our manuscript. We are also thankful to two anonymous reviewers of the previous round for providing valuable comments. We have tried our best to further revise our manuscript based on the additional comments provided by the third reviewer. A point-by-point response to all the comments are available below. The intext additions in the manuscript are available in green font.

Reviewer’s comment

The paper is a good implementation of analysis of financial hardship. The authors did a good analysis of catastrophic expenditure, even comparing budget share approach and CT. The methods used are in line with WHO and World Bank publications on this topic. The results are also in line with what is expected of these types of analysis.

Response: Many thanks for this complement.

The paper could improve if discussions could be deepened. Right now, it basically says households belonging to informal occupations were more prone to catastrophic and impoverishing expenditures. But that finding is well established already in the literature, especially so that informality is very much highly correlated with education level and income levels. The real question is what should be done about it? Given that this is a country level paper, it could have delved deeper into answering this question, but the paper brushed it aside to the limitations of the study, which is quite unfortunate.

Response: Dear Reviewer, many thanks for this observation. As per your advice, we have further elaborated our discussion to include the possible strategies that policy makers can adopt to address the observation that the households belonging to informal occupations were more prone to catastrophic and impoverishing healthcare expenditures. The following sentences are added in the fourth paragraph of the discussion section.

“As a response to this situation, the government of Nepal has launched many social protection schemes focusing particularly towards poor and vulnerable population groups belonging to the informal sector. A few examples are: the implementation of national health insurance, impoverished citizen treatment scheme, and free basic health service package. National health insurance is fully subsidised for identified poor households. However, the result of the poor household identification survey is available for only 26 districts. This has put poor households from remaining districts in a disadvantaged position as they are not being able to benefit from this policy decision. For those households who are enrolled (either through contribution or subsidy) due to inequitable distribution of healthcare providers across provinces, people have to travel to a nearby city or Kathmandu where the cost of treatment could be higher than that could have been in the nearby area. Public facilities in remote areas are understaffed and frequently face a stock out of medicines. Due to this, the people in such areas either have to rely on private clinics and tertiary hospitals in the cities. Due to all these reasons, people from remote and rural areas, who are also from the informal sector, get disproportionately disadvantaged. To tackle this poor households from all the districts should be identified so that they can benefit from public subsidies, healthcare should be more equitably distributed and human resources and logistics should be ensured in the rural and remote areas. The three tiers of government should work together so that the households eligible for health insurance subsidies could be identified and basic health services should be insured at the local level so that people do not have to pay a large sum of money to private and tertiary hospitals even to cure minor ailments.”

For this to be worthy of publication in Plos one, strengthen the abstract and paper to highlight what is new? The disaggregation by occupation is new but then go an extra step and link it to how it can be useful for policy making in Nepal. Cite some initiatives that could be done in Nepal to apply this finding.

Response:

Man thanks for this feedback. We have made the following changes in the manuscript to accommodate your input.

a. We have rephrased the Methods section of the ‘Abstract’ as below in the revised manuscript:

“We measured catastrophic health expenditure (CHE) due to OOP using budget share and capacity-to-pay approach, impoverishment impact at absolute and relative poverty lines. This study is the first of its kind from south-east Asia to analyse disaggregated estimates of financial protection across occupations and geography.”

b. We have added the following sentences in the ‘Materials and method’ section:

“ This study is the first of its kind from south-east Asia to analyse disaggregated estimates of financial protection across occupations and geography using standard methods.”

In addition, we have already highlighted the points mentioned in ‘a’ and ‘b’ in the Discussion section in the original submission as well as in the fourth paragraph of the ‘Introduction’ section.

c. The last paragraph of the ‘Introduction’ section in the original submission provides some background on how this analysis can be useful at policy level. In addition, we have added the following paragraph in the ‘Discussion’ (1st paragraph after the first line) section of revised manuscript to show how disaggregation of financial protection results across occupations could be useful for policymaking in Nepal:

“The disaggregation of results across occupations could be helpful for policymakers in Nepal in the following ways. First, it shows different levels of vulnerability faced by different occupation groups, and thereby identifies the most vulnerable groups that need government intervention. Second, it provides figures to monitor the financial protection of different occupational groups and assess whether public policies are reducing the inequitable distribution of social security measures. Third, it provides a rationale for nationwide risk pooling across occupation groups since the current pooling arrangements through National Health Insurance and Social Security Fund are more focused in the informal sector and formal private sector, respectively.”

In the response to the reviewer comments, the authors were mentioning that Nepal has transitioned to Federal Structure since 2017 and has launched social protection schemes—discuss these in the paper. How does this link to your findings?

Response: We thank the reviewer for this suggestion. We have updated the sixth paragraph of the ‘Discussion’ section by adding the following details:

“The federal structure of the country provides opportunity to provincial and local governments to develop community-based targeting of poor households in the absence of poor-households identification surveys from the federal government. This will help poor households from the rest of the districts benefit from the health insurance subsidy. To improve cross-subsidization health insurance schemes can be made mandatory for both formal and informal sectors of workers. Local governments are in a good position to enforce mandatory enrolment by linking the membership with other social services and benefits provided through the local units (wards and municipalities).”

Specific comments

(1) The first paragraph in the introduction seems misplaced. It talks about Thailand, Turkey, etc. when the paper is about Nepal. It will better to talk about Nepal first to put it in context, before talking about other countries.

Response: Many thanks for this suggestion. Our aim is to analyse financial protection together with standard occupational classification. In order to set a general rationale for such analysis in the global context, we started with the cases of those countries that are making good progress in UHC, and they have tied the healthcare financing arrangements to the different occupations relevant to their economy. Before talking about Nepal, it was necessary to provide an overview of how other countries are reforming their health system and health financing arrangement to minimise the financial burden of their citizens while using healthcare. In Nepal, healthcare financing arrangements appear to cover different occupation groups as well - a further specific case from the global observation. Therefore, in the first paragraph, we started with an overview of financial protection for UHC and its relation with occupational groups in a global context. Then, we proceeded with the case of Nepal including potential relevance to the policy makers.

(2) In the discussion of results, discuss about foregone care. One of the reasons why there might be low catastrophic and impoverishing expenditures among the poor is because they did not seek care, even when they needed to. This is one of the weaknesses of these indicators. If impossible to capture foregone care or unmet need using the survey data, the paper should at least acknowledge that this could exist.

Response: Many thanks for this important concern. We have made following changes in the revised manuscript so that the foregone care is acknowledged.

a. We have modified fourth paragraph of Discussion section as follow:

“These circumstances made healthcare less affordable to them, so the workers belonging to informal sector may have forgone care due to financial barriers. This is one of the reasons for the relatively lower level of CHE in this sub-group.

b. We have added following sentence in the ‘Limitation of this study’

Fourth, because of the data unavailability in the survey, we could not provide an account of foregone care – an important and complementary analysis to the financial protection measures.

(3) Another indicator that was not mentioned in the paper are those who are already poor even before health expenditure, and when they spent for health, they became pushed into poverty even further. As the 2021 global monitoring report of WHO and WB showed, this number is even higher than those who were impoverished due to healthcare. The paper should examine this indicator since they are able to do this with the data they have.

Response: We acknowledge this suggestion. We have included a new table (Table 6) to show the proportion of households pushed further down the poverty across background characteristics and using both the absolute and relative poverty lines. We have added the following paragraph in the ‘Results’ section.

“Table 6 shows the share of population (those already below the poverty line) further pushed into poverty due to healthcare OOP expenditure. At the national level, an estimated 0.38% of the population was pushed further below the extreme poverty line of purchasing-power-parity US $1.90 per capita per day. At US $ 3.2 per capita per day, an estimated 1.25% of the population was pushed further down the poverty line due to healthcare OOP expenditure. Using relative poverty lines of 50% and 60% of median consumption estimated 0.39% and 0.68% of the population, respectively, was further pushed down the poverty due to healthcare OOP. Among household occupation groups, the percentage of the population pushed further down the poverty due to healthcare OOP was least for ‘managers and professionals’ and ‘clerical and support worker’, whereas the percentage was highest for ‘Agriculture’ and ‘Elementary occupations’ at both absolute and relative poverty lines.

Disproportionately greater share of the population from rural areas were pushed further down the poverty compared to the rural area across all the definitions of poverty. Similarly, a larger share of the population from Madhesh, Karnali and Sudurpaschim provinces were disproportionately pushed further down the poverty line compared to other provinces. There was a slight difference in terms of sex of household head, more male headed households were further pushed down the poverty compared to female headed households.

Across the four education categories, the proportion of households pushed further down the poverty was least when the head of household had higher education. Proportion of households pushed further down the poverty increased gradually from least to highest when the household head’s education changed from higher education to less than primary education. Similarly, the share of households pushed further down the poverty increased monotonically from highest wealth quintile to lowest wealth quintile. These observations were consistent with all the definitions of poverty.”

In addition, we have also updated the ‘Discussion’ and ‘Conclusion’ to incorporate the findings shown in Table 6.

Minor points:

(1) In table 1, add a note to say what is the value in parenthesis.

Response: Many thanks for the observation. We have provided a note at the bottom of table 1 and 2.

(2) Typo page 14. “Similar pattern was observed among salaries…” should it be salaried?

Response: Many thanks for the observation. Corrected in the revised manuscript.

(3) Type page 14. There are two periods before “In Gandaki…”

Response: Many thanks for the observation. Corrected in the revised manuscript.

We believe we were able to satisfactorily address all the comments from reviewers and made necessary corrections in the manuscript.

Kind regards,

Authors.

---

## [Editor Report · Decision Letter 2]

10 Jan 2023

Occupational and geographical differentials in financial protection against healthcare out-of-pocket payments in Nepal:  evidence  for universal health coverage

PONE-D-22-17102R2

Dear Dr. Sapkota,

We’re pleased to inform you that your manuscript has been judged scientifically suitable for publication and will be formally accepted for publication once it meets all outstanding technical requirements.

Kind regards,

Kuo-Cherh Huang

Academic Editor

PLOS ONE
---

## [Editor Report · Acceptance letter]

18 Jan 2023

PONE-D-22-17102R2

Occupational and geographical differentials in financial protection against healthcare out-of-pocket payments in Nepal:  evidence for universal health coverage

Dear Dr. Sapkota:

I'm pleased to inform you that your manuscript has been deemed suitable for publication in PLOS ONE. Congratulations! Your manuscript is now with our production department.

Kind regards,

on behalf of

Dr. Kuo-Cherh Huang

Academic Editor

PLOS ONE